# CORE: Concept-Oriented Reinforcement for Bridging the Definition–Application Gap in Mathematical Reasoning

**Zijun Gao**[1]**, Zhikun Xu**[2]**, Xiao Ye**[2]**, Ben Zhou**[2]
[1]University of Illinois Urbana-Champaign   [2]Arizona State University

## Abstract

Large language models (LLMs) often solve challenging math exercises yet fail to apply the concept right when the problem requires genuine understanding. Popular Reinforcement Learning with Verifiable Rewards (RLVR) pipelines reinforce final answers but provide little fine-grained conceptual signal, so models improve at pattern reuse rather than conceptual applications. We introduce CORE (Concept-Oriented REinforcement), an RL training framework that turns explicit concepts into a controllable supervision signal. Starting from a high-quality, low-contamination textbook resource that links verifiable exercises to concise concept descriptions, we run a sanity probe showing LLMs can restate definitions but fail concept-linked quizzes, quantifying the conceptual reasoning gap. CORE then (i) synthesizes concept-aligned quizzes, (ii) injects brief concept snippets during rollouts to elicit concept-primed trajectories, and (iii) reinforces conceptual reasoning via trajectory replacement after group failures, a lightweight forward-KL constraint that aligns unguided with concept-primed policies, or standard GRPO directly on concept-aligned quizzes. Across several models, CORE delivers consistent gains over vanilla and SFT baselines on both in-domain concept–exercise suites and diverse out-of-domain math benchmarks. CORE unifies direct training on concept-aligned quizzes and concept-injected rollouts under outcome regularization. It provides fine-grained conceptual supervision that bridges problem-solving competence and genuine conceptual reasoning, while remaining algorithm- and verifier-agnostic. Related code is available at https://github.com/ARC-ASU/CORE.

## 1 Introduction

Recent LLMs are becoming good at tackling competition-level questions, yet they fall short of conceptual math reasoning beyond applying competition tricks or executing numerical calculations (Yang et al., 2024b; Guo et al., 2025a; Huang & Yang, 2025; Chen et al., 2025). Here, conceptual reasoning means identifying the right concept and applying it in the solution, as opposed to procedural pattern matching often sufficient for GSM8K (Cobbe et al., 2021) or MATH (Hendrycks et al., 2021b) and exposed by perturbation-based tests (Patel et al., 2021; Yu et al., 2024; Mirzadeh et al., 2025; Huang et al., 2025). On many benchmarks, models can mimic solution templates, chain together routine algebraic steps, and even memorize recurring patterns—while still choosing the wrong concept for a problem or failing to correctly apply certain concepts. This gap matters: solving a word problem by spotting a familiar cue is not the same as understanding linear independence, continuity, or convexity and deploying those notions correctly (Li et al., 2025; Huang et al., 2025).

Two factors contribute to the definition–application gap. (i) Exercise-style corpora often reward exploiting surface regularities (formats, keywords, step patterns) rather than applying the intended concepts (Guo et al., 2025b; Wu et al., 2025). (ii) RLVR pipelines (Schulman et al., 2017; Shao et al., 2024) optimize a terminal correctness reward, which is too coarse to teach which concept to invoke or how it should support intermediate steps (Qin et al., 2025). Since concepts require goal-

---

Correspondence to: Zijun Gao (zijung3@illinois.edu) and Zhikun Xu (zhikunxu@asu.edu).

driven instantiation to be testable, we pair concise definitions with aligned quizzes; models readily recite definitions yet often fail the quizzes.

In order to mitigate this definition-application gap, we propose CORE (Concept-Oriented REinforcement), an RL-based framework that turns explicit mathematical concepts into concept-driven training signals in sampling. CORE starts from curating a high-quality data that (i) provide human-verified exercises and (ii) link each exercise to the underlying concept(s), which serve as the in-domain test set and seed data for further generation and training signals. We then expand coverage by generating additional concept-aligned quizzes using LLMs to curate the training set. For the training recipe, CORE has explored three designs: ❶ **Original RL (CORE-Base):** directly training with the generated quizzes by RL algorithms, ❷ **Concept Enhancement (CORE-CR):** injecting concise concept snippets into rollouts to replace part of the original ones when all trajectories are incorrect, and ❸ **KL Divergence (CORE-KL):** implementing the KL divergence term between the concept-guided trajectories and the original ones to implicitly constraint the model towards using concepts. The above design choices investigate in three main components, **data**, **rollouts**, and **loss function**, of RL algorithms to mitigate the definition-application gap and improve the conceptual reasoning. Moreover, this framework wraps around standard policy-gradient reinforcement algorithms without architectural changes. At test time, the trained model is evaluated without providing the concept text, measuring whether concept-aware training translates into genuine conceptual competence rather than reasoning shortcuts.

Empirical results show that CORE delivers consistent and significant improvements across **Qwen2-Math-7B** (Yang et al., 2024a), **DeepSeek-R1-Distill-Qwen-1.5B** (DeepSeek-AI, 2025a), **Qwen2.5-Math-1.5B**(Yang et al., 2024b) and **Llama-3-8B-Instruct** (Team, 2024). Across four model families and a wide range of mathematical reasoning benchmarks, CORE and its variants consistently deliver substantial performance gains, demonstrating strong cross-model generalization. On Qwen2-Math-7B, CORE variants achieve large improvements, with gains of up to **+9.3%** on TEXTBOOK and **+9.6%** on THEOREMQA, indicating enhanced conceptual alignment and deeper reasoning. For DeepSeek-R1-DQ-1.5B, CORE-CR improves performance by **+1.3%** on MMLU-STEM and **+1.2%** on SVAMP. On Qwen2.5-Math-1.5B, CORE yields consistent gains, boosting MINERVA MATH by **+3.3%** and TABMWP by **+1.9%**. Finally, on Llama-3-8B-Instruct, CORE-CR outperforms the Vanilla baseline, achieving up to **+3.3%** on TABMWP, while also improving results on MMLU-STEM and SVAMP. These results demonstrate that, without any architectural modifications, CORE consistently enhances conceptual understanding and reasoning ability through explicit concept injection and concept-aware optimization.

## 2    RELATED WORKS

**Mathematical Reasoning in LLMs**    Recent systems approach math reasoning through specialized training or sheer scale. WizardMath (Luo et al., 2025) uses reinforcement learning from Evol-Instruct feedback. MAmmoTH (Yue et al., 2024) blends chain-of-thought and program-of-thought for hybrid tuning. Qwen2.5-Math (Yang et al., 2024b) continues from a general model on a large curated math corpus. Llemma (Azerbayev et al., 2024) is pre-trained on Proof-Pile. DeepSeekMath (Shao et al., 2024) adds about 120B math tokens and introduces *Group Relative Policy Optimization* (GRPO). InternLM2-Math (Ying et al., 2024) unifies chain-of-thought, reward modeling, and formal reasoning. General-purpose models also lean heavily into math: Llama 3.1 (Team, 2024) up-samples math data, DeepSeek-R1 and DeepSeek-V3 (DeepSeek-AI, 2025b) raise math and programming proportions across trillions of tokens, Claude 3.7 (Anthropic, 2025) emphasizes transparent multi-step reasoning, Gemini (Gemini, 2025) targets long-chain deduction, and OpenAI o1 and o3 (OpenAI, 2024; 2025) scale test-time compute. Despite these advances, many pipelines reward final answers or rely on data scale, leaving concept selection and application under-taught. Our CORE addresses this by injecting explicit concept signals into rollouts and regularizing outcomes, yielding consistent gains on concept-dependent evaluations while remaining agnostic to the underlying RL algorithm.

**Mathematical Benchmarks**    A wide range of math benchmarks now probes reasoning at various levels. At the elementary level, GSM8k is the standard for multi-step math problems. MAWPS (Koncel-Kedziorski et al., 2016) aggregates sources, ASDiv (Miao et al., 2021) adds type and grade labels, and SVAMP (Patel et al., 2021) stresses robustness through controlled perturbations.

Cross-lingual coverage includes CMATH for Chinese primary school (Wei et al., 2023) and CN Middle School 24, while standardized suites such as Gaokao 2023 EN, SAT Math, and MMLU-STEM enable broader STEM-wide comparisons (Zhong et al., 2023; Hendrycks et al., 2021b). For competition-level reasoning, MATH curates Olympiad and contest problems with stepwise solutions, and OlympiadBench (He et al., 2024) extends to bilingual and multimodal settings that emphasize proof-style reasoning and reduce contamination risk. Our evaluations of CORE are based on in-domain concept–exercise suites and out-of-domain math benchmarks, showing consistent gains over strong baselines and highlighting improvements specifically on concept-dependent categories.

**Conceptual Reasoning**   Answer accuracy alone doesn't reveal whether the right concepts were selected and used, several works probe whether models actually select and use the right concepts. Specifically in math, conceptual reasoning requires people to reason around math concepts and axioms at the play of math hypothesis, statements and problems (Simon, 2011). THEOREMQA (Chen et al., 2023) targets theorem application across STEM, explicitly requiring mapping from a named theorem to its correct use in problem solving. GSM-SYMBOLIC (Mirzadeh et al., 2025) examines symbolic generalization limits of models trained on GSM8K-style data. COUNTERMATH (Li et al., 2025) proposes counterexample-driven, concept-sensitive evaluations to diagnose superficial cues versus true concept use. Complementary directions include a conceptualization framework that maps abstract questions into verifiable symbolic programs (Zhou et al., 2024), a self-supervised analogical learning scheme that transfers high-level solutions across cases (Zhou et al., 2025), and a Bayesian inference formulation coupling abductive proposals with structured deduction for calibrated decisions (Feng et al., 2025). Overall, these efforts indicate that a single end-point score is insufficient for assessing whether models select and correctly apply concepts; concept-aligned training signals or structured evaluation protocols are required.

## 3   CORE: CONCEPT-ORIENTED REINFORCEMENT LEARNING

### 3.1   OVERVIEW

We study math reasoning where success depends mainly on conceptual math reasoning rather than replaying surface templates in the training data. Our framework CORE proceeds through **dataset curation**, **gap diagnostics**, and **concept reinforcement**. For *dataset curation*, we have leveraged a classical mathematical textbook with clear associations between concepts and exercises for training and evaluation. For *gap diagnostics*, we have used the curated data to both qualitatively and quantitatively identify the definition-application gaps in the conceptual mathematical reasoning. For *concept reinforcement recipe*, we have mainly designed three training recipes in reinforcing the models' conceptual mathematical reasoning.

### 3.2   DATASET CURATION

To acquire rigorous conceptual reasoning signals, we curated a corpus from a canonical textbook, *Advanced Algebra* (3rd Edition) (Yao & Xie, 2015). This source was chosen for two-fold reasons. First, it provides a comprehensive and structured curriculum in linear algebra, progressing logically from foundational concepts like determinants and matrices to advanced topics such as linear spaces and canonical forms. Its ten chapters are methodically structured, each containing: i) core concept definitions ($\mathcal{C}$), ii) illustrative examples, and iii) concept-aligned exercises ($\mathcal{E}$), where exercises in a given chapter primarily test the concepts introduced in that same chapter. The textbook's long-standing use and human verification ensure a logical progression of topics and coherent conceptual dependencies, making it an ideal corpus for developing and evaluating a concept-oriented learning paradigm. Second, by manually translating this Chinese textbook into English, we significantly mitigate the risk of training data contamination present in many existing English-language corpora.

The extraction yielded 236 concept texts, 703 examples, and 140 multiple-choice questions sourced from the exercise sections. More details are provided in Appendix A.

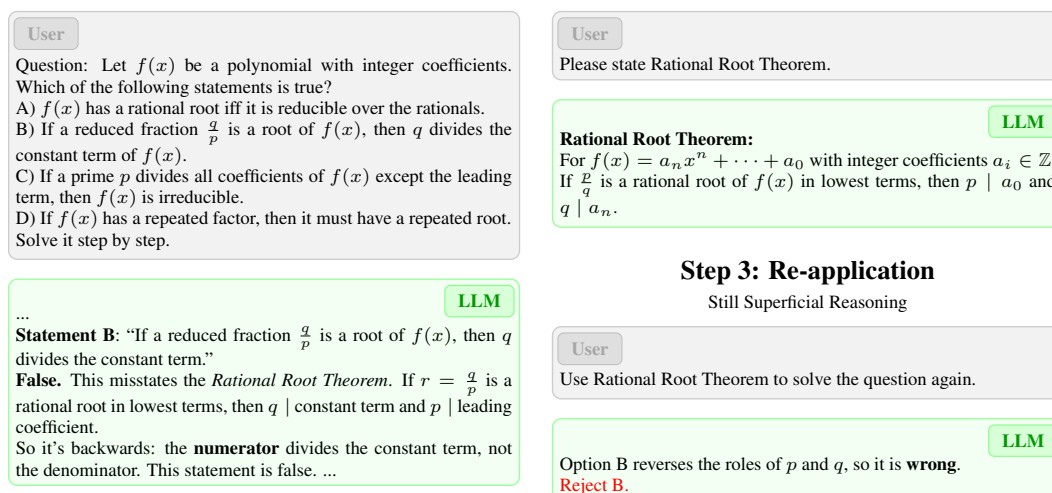

Figure 1: An example of ChatGPT-4o's superficial understanding of the Rational Root Theorem. Although the model correctly recalls the theorem, it fails to engage in structural verification across reasoning steps.

## 3.3 GAP DIAGNOSTICS

### 3.3.1 PROBING THE GAP BETWEEN KNOWLEDGE RECITATION AND APPLICATION

The structured nature of our curated corpus, with its explicit mapping between concepts ($\mathcal{C}$) and exercises ($\mathcal{E}$), provides an ideal testbed to diagnose a critical failure mode in state-of-the-art LLMs: the disconnect between parametric knowledge and its application in problem-solving. We conduct the following sanity check to probe this gap, which provides the foundational motivation for our proposed training framework.

We conduct a qualitative analysis of multiple-choice exercises that a strong baseline, GPT-4o,[1] fails due to conceptual errors. After an incorrect answer, we prompt the model to describe the core concept underlying the problem (e.g., "Describe linear independence."). In most cases, the model correctly recites the definition and key properties, suggesting that the knowledge is parametrically stored. Figure 1 shows a representative case: GPT-4o accurately states the Rational Root Theorem but still misapplies it when the numerator and denominator are swapped. This reveals a definition–application gap: the model can recall a concept yet fails to deploy it flexibly in reasoning.

### 3.3.2 SYNTHETIC CONCEPT PROBES AND ROBUSTNESS EVALUATION

**Measuring Conceptual Understanding via Concept Probes.**   A fundamental challenge in evaluating mathematical reasoning is the difficulty of quantitatively measuring a model's grasp of specific concepts. High-level benchmark scores often obscure fine-grained conceptual failures. The highly structured nature of our curated textbook, however, provides a unique opportunity to address this challenge. To create a direct and quantifiable measure of conceptual understanding, we introduce the idea of **Concept Probes**: targeted quizzes generated directly from the textbook's concept definitions and theorems, where a model's performance on these probes serves as a proxy for its mastery of the underlying concepts.

To realize this, we constructed a new dataset of conceptual quizzes. This process involved two key stages: generation and validation.

1. **Generation:** We prompted a powerful generator model, Qwen2.5-72B-Instruct, to create 5–8 multiple-choice quizzes for each of the 236 concept texts in our corpus. This resulted

---

[1]https://openai.com/index/hello-gpt-4o/

in a candidate pool of 1,200 quizzes, each designed to be closely tied to its source concept and formatted with standard LaTeX.

2. **Validation:** To ensure the quality and validity of these synthetic quizzes, we designed a rigorous filtering pipeline using a separate, powerful assessor model, GPT-4o. This cross-model validation strategy is intentionally designed to reduce harvester bias. For each quiz, GPT-4o evaluated six dimensions (e.g., clarity, correctness, uniqueness) and provided an overall rating and a confidence level. We discarded the 90 quizzes that were rated "Fair" or "Poor" with high confidence. This stringent process yielded a final set of **1,110 high-quality quizzes** that serve as our Concept Probes.

**Diagnostic Experiment: Robustness to Superficial Perturbations.** To validate our hypothesis that models rely on superficial heuristics, we designed a diagnostic experiment to test the robustness of their conceptual knowledge. Using our 1,110 curated quizzes, we use a **Robust Evaluation** protocol. For each quiz, we generate three variants by randomly permuting the order of its multiple-choice options. A model is considered to have *robustly* solved a problem **only if** it correctly answers the original question *and* all three of its permuted variants. This protocol is designed to test whether a model's understanding is invariant to semantically-irrelevant changes that preserve the core concept.

We applied both the standard and our Robust Evaluation protocols to a suite of contemporary models, including Qwen2-Math-7B, OLMo-2-7B (OLMo et al., 2025), and Llama-3-8B. The results, illustrated in Table 1, reveal a stark and consistent performance gap across all models. For instance, while a model like OLMo-2-7B may achieve high accuracy (e.g., $> 70\%$) under the standard protocol, its performance plummets to below $50\%$ under Robust Evaluation. This significant degradation provides strong empirical evidence for our hypothesis, demonstrating that the models' success is heavily reliant on shallow heuristics rather than a deep, structural understanding of the underlying concepts.

Table 1: Performance comparison under Original vs. Robust Evaluation protocol across different models. The best performance in each column is highlighted in bold.

| Model | Standard evaluation | | Robust evaluation | |
|---|---|---|---|---|
| | pass@1 accuracy | self-consistent | pass@1 accuracy | self-consistent |
| Qwen-2-Math-7B | 74.33% +0 | 87.0% +0 | 45.92% -28.4 | 76.0% -11.0 |
| OLMo-2-7B | 57.83% +0 | 70.17% +0 | 36.25% -21.6 | 44.42% -25.8 |
| LLaMA-3-8B | 44.75% +0 | 70.92% +0 | 20.25% -24.5 | 46.75% -24.2 |

### 3.4 CONCEPT REINFORCEMENT RECIPE

To bridge the gap between procedural mimicry and conceptual understanding, we propose CORE, illustrated in Figure 2. CORE is an RL-based framework designed to inject conceptual knowledge into models through any policy gradient based RL algorithm. The core idea is to conditionally intervene during training with concept-guided instruction precisely when the model demonstrates a failure in understanding, guiding the policy update towards a more robust, concept-grounded reasoning process. Our CORE framework mainly consists of the following three design choices, instantiated with GRPO as the standard RL backbone in this paper. For completeness, we also discuss and evaluate a PPO-based variant in Appendix C.1.

**CORE-Base: Standard RL on Conceptual Quizzes.** The foundational approach within our framework, **CORE-Base**, involves training the policy $\pi_\theta$ directly on our curated set of conceptual quizzes ($\mathcal{Q}$) using the standard GRPO algorithm. In this setting, the model learns from the conceptual data without any further explicit guidance during the training process. This approach measures the model's ability to implicitly learn concepts from the rich question-answer pairs generated from concepts.

**CORE-CR: *C*oncept-Guided Trajectory *R*eplacement.** Building upon the base setting, **CORE-CR** introduces a conditional intervention triggered by a *conceptual failure event* (i.e., all $N$ responses in a GRPO group are incorrect). Upon triggering, we form a concept-guided prompt

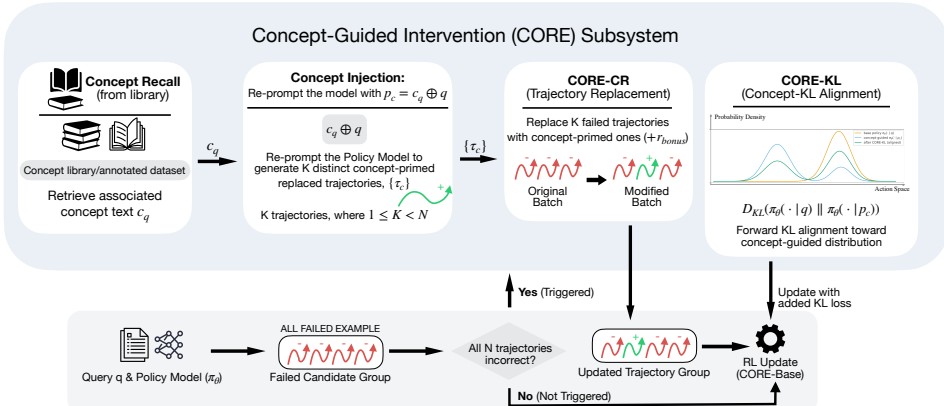

Figure 2: Overview of the Concept-Guided Reinforcement (CORE) framework. For a given query, the policy model generates multiple candidate solutions. If any solution is correct, CORE-Base proceeds. When all solutions fail, CORE activates concept-guided correction: the Concept Recall module retrieves relevant domain knowledge, and Concept Injection re-prompts the model with this guidance to form corrected trajectories. CORE-CR replaces failed paths with these concept-grounded ones to recover the learning signal, while CORE-KL distills the concept-enhanced trajectories into the base policy via a forward KL loss.

$p_c = c_q \oplus q$ where $q$ is the original problem from our quiz dataset, $c_q$ is its associated ground-truth concept text, and $\oplus$ denotes concatenation. We then generate $K$ new trajectories $\{\tau_{c,1}, \ldots, \tau_{c,K}\}$ from the concept-guided policy, where $1 \leq K < N$. We then **randomly select and replace** $K$ trajectories from the original failed group with these new concept-guided ones. To incentivize learning with concepts, we assign the new trajectories an augmented reward:

$$R'(\tau_{c,j}) = R(\tau_{c,j}) + r_{\text{bonus}}$$

where $r_{\text{bonus}} > 0$ is a hyperparameter. The GRPO update is then performed on this partially replaced, concept-guided batch. Notably, there is a recent work called BREAD (Zhang et al., 2025), which shares a very similar methodology to CORE-CR while derives from rethinking the advantages of SFT and RL instead of improving conceptual reasoning. However, as empirically demonstrated in Section 5.4, CORE achieves significant gains without distilling reasoning signals from a larger teacher model. This autonomy distinguishes CORE from mainstream "expert-anchored" approaches like BREAD, which typically require superior external models to provide trajectory guidance.

**CORE-KL: Concept-Guided *KL*-Regularization.** The **CORE-KL** method introduces a fine-grained regularization signal to guide the policy's internal reasoning process. This approach is also triggered by a *conceptual failure event*. Instead of directly replacing trajectories, this method encourages the model's standard step-by-step predictive process at each timestep $t$, denoted $\pi_\theta(\cdot \mid q, y_{<t})$, to align with the more robust process it exhibits when primed with a concept, $\pi_\theta(\cdot \mid p_c, y_{<t})$.

We formulate this as a **forward KL-divergence** objective. This choice is deliberate: it encourages the base policy to cover the full distribution of reasoning paths considered by the concept-guided "teacher" policy, rather than collapsing to a single mode, fostering a more comprehensive distillation of the entire reasoning process. Upon a conceptual failure trigger, we first sample a high-quality reference trajectory, $Y^* = (y_1^*, \ldots, y_T^*)$, from the *current, online concept-guided policy*, i.e., $Y^* \sim \pi_\theta(\cdot \mid p_c)$. Our objective is then to minimize the KL-divergence between the next-token predictive distributions of the guided and un-guided policies at each timestep $t$, conditioned on the prefix of the reference trajectory $Y_{<t}^*$. This is formulated as a loss term added to the base RL objective:

$$\mathcal{L}_{\text{total}} = \mathcal{L}_{\text{GRPO}} + \lambda_{\text{KL}} \mathbb{E}_{Y^* \sim \pi_\theta(\cdot|p_c)} \left[ \sum_{t=1}^{|Y^*|} D_{\text{KL}} \Big( \pi_\theta(\cdot \mid p_c, y_{<t}^*) \parallel \pi_\theta(\cdot \mid q, y_{<t}^*) \Big) \right]. \quad (1)$$

where $\pi_\theta(\cdot \mid \text{context}, y_{<t}^*)$ is the current policy's probability distribution over the next token. This forces the model's internal reasoning process on the original problem $q$ to faithfully mimic the process it would follow if it were explicitly given the concept $c_q$.

CORE-Base primarily functions as a consolidation mechanism, reinforcing the application of concepts already encountered during pre-training, whereas CORE-CR and CORE-KL serve as complementary corrective strategies for concepts that are not yet reliably mastered, operating in parallel through explicit and implicit forms of concept intervention. Together, these variants constitute parallel and complementary ways of incorporating conceptual signals into reinforcement learning within a unified framework.

# 4 EXPERIMENTS

## 4.1 BASELINE MODELS

We select **Qwen2-Math-7B** as our primary evaluation model. This choice is motivated by its moderate mathematical proficiency: it demonstrates non-trivial reasoning skills while still leaving sufficient headroom on our quiz tasks, thus offering a reliable foundation for assessing the impact of CORE. For this model, we report results for all three CORE variants (CORE-Base, CORE-CR, and CORE-KL). Beyond this, we further evaluate CORE-CR on **DeepSeek-R1-Distill-Qwen-1.5B**, **Qwen2.5-Math-1.5B**, and **Llama-3-8B-Instruct**, highlighting the algorithm's robustness and cross-model generalization across both math-specialized and instruction-tuned settings.

## 4.2 TRAINING SETTINGS

Our CORE variants introduce method-specific hyperparameters: a reward bonus $r_{bonus}$ for CORE-CR and a dynamic KL coefficient $\lambda_{KL}$ for CORE-KL. All training configurations and extensive hyperparameter details are deferred to Appendix B.1.

## 4.3 EVALUATION SETTINGS

**In-domain Test.**   To measure in-domain performance, we use the 140 multiple-choice exercises curated from the textbook. These exercises serve as a high-quality and reliable measure of concept application due to their expert authorship and direct alignment with the textbook's definitions. We denote this test set as **Textbook**.

**Out-of-domain Benchmarks.**   To assess whether the conceptual understanding fostered by CORE generalizes beyond our curated training data, we evaluate our trained models on a diverse suite of out-of-domain benchmarks. These benchmarks were specifically chosen to probe for different facets of mathematical reasoning, from multi-step arithmetic and competition math to robustness against perturbations. This evaluation is critical to demonstrate that CORE does not simply overfit to the textbook's style, but rather instills a more fundamental and transferable reasoning capability.

We evaluate models trained with three instantiations of our CORE framework—CORE-Base, CORE-CR, and CORE-KL—on the following out-of-distribution benchmarks: **GSM8K** (Cobbe et al., 2021), **ASDiv** (Miao et al., 2021), **MAWPS** (Koncel-Kedziorski et al., 2016), **TabMWP** (Lu et al., 2023), **MATH** (Hendrycks et al., 2021b), **MMLU-STEM** (Hendrycks et al., 2021a), **Gaokao 2023 (EN)** (Zhong et al., 2023), **Gaokao-Math-QA** (Zhong et al., 2023), **CMATH** (Wei et al., 2023), **Minerva Math** (Lewkowycz et al., 2022), **SVAMP** (Patel et al., 2021), **CounterMath** (Li et al., 2025), **TheoremQA** (Chen et al., 2023), and **OlympiadBench** (He et al., 2024). A detailed description of the datasets is provided in Appendix B.2.

**Metrics.**   All results are reported with self-consistency (SC@21, $T = 0.7$); see Appendix B.3.

As shown in Table 2, models trained with the CORE framework exhibit consistent and significant performance improvements across the majority of out-of-domain benchmarks when compared to the vanilla baseline. This demonstrates that CORE successfully enhances the models' underlying reasoning abilities in a way that generalizes to unseen problem distributions and formats.

Table 2: Main table of accuracy (%) under SC@21 ($T$=0.7). Columns use *two-letter* abbreviations: **TB**=Textbook, **GS**=GSM8K, **AD**=ASDiv, **MW**=MAWPS, **TM**=TabMWP, **MH**=MATH, **MS**=MMLU-STEM, **GK**=Gaokao 2023 (EN), **CM**=CounterMath (reported as F1), **TQ**=TheoremQA, **OL**=OlympiadBench.

| Model | Method | TB | GS | AD | MW | TM | MH | MS | GK | CM | TQ | OL |
|---|---|---|---|---|---|---|---|---|---|---|---|---|
| | Vanilla | 46.4 | 89.8 | 95.1 | 96.8 | 90.2 | 69.1 | 72.9 | 55.3 | 13.2 | 34.6 | 28.7 |
| | SFT | 45.0 | 86.6 | 94.1 | 96.6 | 85.6 | 59.4 | 72.4 | 46.5 | **16.7** | **44.2** | 19.7 |
| Qwen2-Math-7B | CORE-Base | 50.7 | 90.8 | 95.4 | 97.2 | 92.6 | 71.1 | 72.9 | **59.5** | 13.5 | 40.4 | 33.9 |
| | CORE-CR | 52.1 | **91.1** | **95.7** | 97.3 | **93.6** | **71.4** | 72.6 | 58.4 | 15.5 | 42.3 | **34.5** |
| | CORE-KL | **55.7** | 90.7 | 95.5 | **97.5** | 90.6 | 70.5 | **73.1** | **59.5** | 15.8 | **44.2** | 32.9 |

Table 3: Results on the diagnostic subset $W$ ($|W| = 19$). A problem is Concept-Selection iff both CORE-CR and CORE-KL explicitly invoke the target concept and show no heuristic cues; Heuristic-selection iff both rely on heuristics with no substantive concept use; otherwise Mixed.

| Category | # Problems | Probability(%) |
|---|---|---|
| Concept-selection | 10 | 52.6 |
| Mixed | 9 | 47.4 |
| Heuristic-selection | 0 | 0.0 |
| **Total** | **19** | **100.0** |

# 5 ANALYSIS

## 5.1 DOES CORE ENHANCE CONCEPT SELECTION AND APPLICATION?

We first verify what training with CORE actually changed: are the observed accuracy gains attributable to improved *concept selection and application*, rather than superficial heuristics, and achieved without any test-time concept prompting? To probe this, we evaluate four model variants (Vanilla, CORE-Base, CORE-CR, and CORE-KL) on 140 textbook exercises, and define a diagnostic subset $W$ comprising problems that are solved by both CORE-CR and CORE-KL but not solved by either Vanilla or CORE-Base,

$$W = \{\, i \mid (\text{Vanilla fails on } i \text{ or CORE-Base fails on } i) \wedge (\text{CORE-CR and CORE-KL succeed on } i) \,\}.$$

yielding $|W| = 19$ (12 Vanilla-only failures, 4 CORE-Base-only failures, 3 shared failures). For each $i \in W$, we read the generations from CORE-CR and CORE-KL and score along two dimensions: (i) **concept hits**—the output explicitly mentions the task's target concept and uses it correctly in the reasoning; and (ii) **heuristic cues**—guessing, option elimination without justification, surface pattern matching, or plug-in substitution without conceptual warrant.

For labeling, we keep the rules simple. A problem is called Concept-Selection if both CORE outputs (CORE-CR and CORE-KL) contain a concept hit and neither shows heuristic cues; it is classified as Heuristic-Selection if both outputs rely on heuristics and contain no concept hit; otherwise, it is categorized as Concept+Heuristic (mixed). We require "two hits" (one per CORE variant) so that if either variant omits the concept, the instance is not counted as Concept-Selection. As Table 3 shows, 10/19 (52.6%) cases are Concept-Selection , 9/19 (47.4%) are Mixed, and 0/19 are Heuristic-selection. Due to our strict constraints, it rules out superficial shortcutting as the primary driver of the gains. On one representative question (see Table 9), even after appending a targeted concept prompt at test time, CORE-Base remained incorrect, whereas CORE-KL solved it without any prompt. While this is a single illustrative case, it reinforces that the observed gains come from training-induced mechanism change rather than prompt engineering. Taken together, the evidence indicates a mechanism shift: CORE improves accuracy mainly by strengthening concept selection and application.

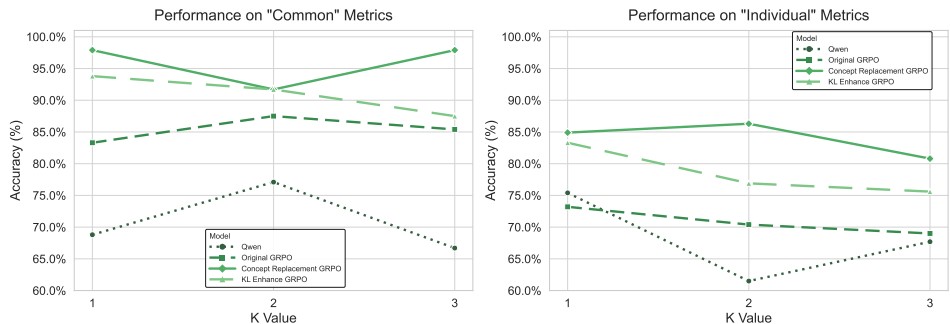

Figure 3: Performance comparison on Common vs Individual metrics.

## 5.2 DOES CORE IMPROVE ROBUSTNESS TO IRRELEVANT CONCEPT PERTURBATIONS?

We next ask whether training with CORE yields improved *robustness* to irrelevant concept cues. Using 140 high-quality, thematically related textbook exercises, we prepend $K \in \{1, 2, 3\}$ concepts that are not directly related to the target concept to each question and measure whether the model can still retain the correct answer under such perturbations. To ensure that distractors are not directly related, we select concepts drawn from different textbook chapters. For each question and each $K$, we sample one fixed distractor set once (single random seed) and use the same set for all models to enable paired comparisons.

To quantify retention under perturbation, we report **RUD**$_K$ (Retention Under Distractors): accuracy on perturbed items restricted to questions a model already solved without perturbation. Formally, letting $S_m$ be the questions solved by model $m$ in the unperturbed setting, $x_i^{(K)}$ be the question with $K$ distractors prepended and $y_i$ be the correct label,

$$\mathrm{RUD}_K(m) \;=\; \frac{1}{|S_m|} \sum_{i \in S_m} \mathbf{1}\Big\{ m\Big(x_i^{(K)}\Big) = y_i \Big\}.$$

We evaluate four models—Vanilla, CORE-Base, CORE-CR, and CORE-KL—under two splits: **Common** (items solved by all models; $n = 48$) and **Individual** (per-model solved sets: Vanilla 65 / CORE-Base 71 / CORE-CR 73 / CORE-KL 78). The resulting $\mathrm{RUD}_K$ curves for each split can be seen in Figure 3.

As $K$ increases, models trained with CORE show consistently smaller accuracy drops than Vanilla and CORE-Base on both splits, with the CORE-CR variant particularly robust. This pattern indicates that CORE not only improves the accuracy of the headline, but also improves the robustness, making predictions more stable against perturbations of irrelevant concepts.

## 5.3 DOES CORE APPLY TO BASE AND INSTRUCTION-TUNED MODELS?

We ask whether these gains persist across both base and instruction-tuned models. We therefore apply CORE-CR to three representative models: **DeepSeek-R1-Distill-Qwen-1.5B**, **Qwen2.5-Math-1.5B**, and **Llama-3-8B-Instruct**. We evaluate them under the same SC@21 ($T$=0.7) protocol across diverse out-of-domain suites (Table 4; abbreviations follow the table).

CORE-CR yields consistent **average** improvements across all three models: DeepSeek-R1-DQ-1.5B improves from $72.7 \rightarrow 73.1$ (+0.4), Qwen2.5-Math-1.5B from $72.1 \rightarrow 72.4$ (+0.3), and Llama-3-8B-Inst from $58.1 \rightarrow 58.9$ (+0.8). Taken together, these results indicate that CORE is model-agnostic, providing consistent gains across both base and instruction-tuned models.

## 5.4 IS CORE DRIVEN BY KNOWLEDGE DISTILLATION OR INTRINSIC CONCEPTUAL REINFORCEMENT?

A critical concern in RL training with synthetic data is whether the observed gains stem from simple **knowledge distillation** from a superior teacher model. To disentangle CORE's mechanism

Table 4: DeepSeek-R1-Distill-Qwen-1.5B, Qwen2.5-Math-1.5B, and Llama-3-8B-Instruct: Out-of-domain benchmark accuracy (%) under SC@21 ($T$=0.7). Columns use *two-letter* abbreviations: **CA**=CMath, **GQ** = GaokaoMathQA, **GK**=Gaokao 2023 (EN), **MH**=MATH, **MW**=MAWPS, **MM**=Minerva Math, **MS**=MMLU-STEM, **SV**=SVAMP, **TM**=TabMWP

| Model | Method | CA | GQ | GK | MH | MW | MM | MS | SV | TM |
|---|---|---|---|---|---|---|---|---|---|---|
| DeepSeek-R1-DQ-1.5B | Vanilla | 90.8 | 75.2 | 58.2 | 68.6 | 96.9 | 23.9 | 58.6 | 92.8 | **89** |
| | CORE-CR | **91.5** | **75.5** | **59.2** | **69** | **97.1** | **24.3** | **59.9** | **94** | 87.6 |
| Qwen2.5-Math-1.5B | Vanilla | **91** | **60.7** | 59.5 | 75.9 | 97.1 | 26.1 | **61.2** | 93 | 84 |
| | CORE-CR | **91** | 57.8 | **60** | **77.2** | **97.6** | **29.4** | 59.4 | **93.3** | **85.9** |
| Llama-3-8B-Inst | Vanilla | 78.8 | 25.9 | 35.8 | **41.6** | 93.8 | **16.9** | 63.2 | 90 | 77.1 |
| | CORE-CR | **79.7** | **26.2** | **36.6** | 39.9 | **95.4** | 15.8 | **64.6** | **91.6** | **80.4** |

Table 5: Results of the Self-Supervised experiment using **Qwen2-Math-7B-Instruct** as the generator and **Qwen2-Math-7B** as the learner. Abbreviations: **AD**=ASDIV, **GK**=GAOKAO 2023 (EN), **GS**=GSM8K, **MH**=MATH, **MS**=MMLU-STEM, **MW**=MAWPS, **OL**=OLYMPIADBENCH, **TM**=TABMWP.

| Benchmark | GS | AD | MW | TM | MH | MS | GK | OL |
|---|---|---|---|---|---|---|---|---|
| Vanilla | 89.8 | 95.1 | 96.8 | 90.2 | 69.1 | 72.9 | 55.3 | 28.7 |
| CORE-CR | **91.4** | **95.5** | **97.6** | **92.5** | **70.4** | **73.1** | **57.7** | **32.9** |

from such effects, we conducted a "Self-Supervised" experiment by restricting the entire pipeline to the **Qwen2-Math-7B** family, thereby eliminating any external "expert" guidance from ultra-large LLMs. Specifically, we utilized **Qwen2-Math-7B-Instruct** to generate 885 quizzes based on the 236 textbook concepts. To ensure quality without external supervision, we employed a self-verification protocol where only the 670 quizzes that the model could solve correctly via self-consistency (SC@21) were retained.

As shown in Table 5, CORE-CR delivers consistent and substantial improvements across benchmarks even with self-generated and potentially noisy concept probes. This evidence suggests that CORE's efficacy does not depend on perfectly clean quiz data or a superior expert model. Instead, the framework is **naturally robust**, capable of extracting useful learning signals through its intrinsic intervention logic, effectively bridging the reasoning gap through a closed-loop, self-improving process. To further disentangle whether the observed performance gains arise from intrinsic conceptual reinforcement rather than reinforcement learning artifacts or passive process supervision, we provide extensive ablation studies in Appendix C.2.

## 6 CONCLUSION

In this work, we introduced **CORE (Concept-Oriented REinforcement)**, a reinforcement learning framework designed to bridge the gap between conceptual definition and practical application in mathematical reasoning. By leveraging a curated concept–exercise corpus, we diagnosed the limitations of current LLMs and injected explicit concept signals into the training process. Specifically, through concept-aligned quizzes, concept-guided trajectory replacement, and KL-based divergence regularization, CORE delivers granular supervision that extends beyond simple outcome correctness. Extensive experiments on both base and instruction-tuned models demonstrate that CORE consistently yields substantial performance gains across in-domain and out-of-domain benchmarks, particularly in conceptual precision and robustness. Crucially, these improvements are achieved without architectural modifications, ensuring seamless compatibility with standard policy-gradient methods. Our findings underscore that grounding reinforcement learning in explicit concepts can transcend surface-level heuristics, moving LLMs toward genuine conceptual competence. We hope this work motivates further exploration of concept-centered training signals in mathematics and other domains where principled reasoning is paramount.

ETHICS STATEMENT

Our dataset is curated from high-quality educational resources originally published in Chinese. We contacted the primary author, who indicated they could not grant permission at this time due to unclear regulations around LLM training and evaluation in Copyright Law of China. After consulting legal guidance, we understand that limited use of such materials for non-commercial academic research may be permissible. Accordingly, our use is strictly for research and education; we do not redistribute substantial verbatim text. The research artifacts (codes, prompts, scripts, structured concept–exercise mappings, and model-generated quizzes/snippets) are derived and only small illustrative samples, that does not contain substantial portions of the original expression, would be presented. We cite sources and will promptly honor takedown or correction requests. Any released artifacts are for research use only and may not be used commercially; parties seeking commercial use should contact the rights holders. This statement is not legal advice, and we will adjust our practices as regulations evolve.

REPRODUCIBILITY STATEMENT

Our proposed framework is specified in §3. Our data curation, motivation verification, and training recipes are illustraed under it with subsections named as *Dataset Curation*, *Gap Diagnostics*, and *Concept Reinforcement Recipe*. The training and evaluation settings appear in §4 and Appendix B. We will provide a full code repo with codes, configs, and scripts to run training and evaluation end-to-end in the camera-ready version.

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

# APPENDIX

## A DETAILS FOR TEXTBOOK DATA

### A.1 DATA CURATION DETAILS

Our data curation followed a multi-stage pipeline to ensure high fidelity. We first employed an OCR tool[2] to digitize the textbook. The concept and exercise sections then underwent a manual verification stage, with any recognition errors corrected using GPT-4o. Subsequently, the entire Chinese corpus was translated into English via GPT-4o, followed by another round of human verification on the key sections to ensure accuracy.

## B EXPERIMENT SETUP

### B.1 MORE TRAINING DETAILS

In the main experiments, we train models for 3 epochs on four H200 GPUs with GRPO by the *verl* (Sheng et al., 2024) framework. In our experiments, we set $r_{bonus} = 0.4$ for **CORE-CR**. For the **CORE-KL**, we use $\lambda_{KL} = 0.03$ when the reference concept-guided trajectory is correct, and $\lambda_{KL} = 0.005$ otherwise. The policy is optimized using Adam with an actor learning rate of $1 \times 10^{-6}$, and a standard KL-divergence penalty with a fixed coefficient of 0.001 is applied against the reference policy. During training, we set the sampling temperature to 0.7, with a batch size of 128 and a mini-batch size of 32. For each prompt, *four* responses are generated to conduct GRPO updates. Rewards follow a *binary* scheme (1 for correct, 0 for incorrect). The maximum prompt length is capped at 1024, while the maximum response length varies across models, being set to 1024 for Qwen2-Math-7B, 2048 for Qwen2.5-Math-1.5B and Llama-3-8B-Instruct, and 6000 for DeepSeek-R1-Distill-Qwen-1.5B to accommodate their extended reasoning contexts.

### B.2 DETAILS OF EVAULATION DATASETS

All evaluation datasets are sourced from the Qwen2.5-Math evaluation repository[3]. Their details are summarized below.

GRADE SCHOOL & MIDDLE SCHOOL LEVEL

- GSM8k: A dataset of approximately 8,500 high-quality, linguistically diverse elementary school math word problems. These problems require 2 to 8 steps of reasoning to solve and primarily involve basic arithmetic operations ($+, -, \times, \div$). The purpose of this dataset is to evaluate a model's ability to perform multi-step mathematical reasoning.

- ASDiv: An English math word problem dataset that is diverse in both linguistic patterns and problem types. It aims to comprehensively evaluate the true capabilities of math problem solvers, preventing models from achieving high scores merely by "memorizing" solutions to similar problems. Each problem is annotated with its problem type and grade level.

- MAWPS: A collection of several thousand English math word problems sourced from various online educational websites. Its goal is to provide an extensible repository of math problems for researchers to use and expand upon, covering a variety of basic arithmetic and algebraic problems.

- TabMWP: A large-scale dataset containing over 38,000 math word problems, distinguished by the inclusion of a table as context for each problem. To solve these, a model must be able to retrieve, integrate, and perform multi-step mathematical reasoning on information from both textual and tabular sources.

- CMath: A Chinese-language dataset designed to evaluate language models on elementary school mathematics. It contains carefully curated word problems covering fundamental

---

[2] https://github.com/opendatalab/MinerU
[3] https://github.com/QwenLM/Qwen2.5-Math

arithmetic operations (addition, subtraction, multiplication, and division) and simple logic-based reasoning. The dataset challenges models to perform precise quantitative reasoning in Chinese, testing both mathematical competence and cross-lingual generalization.

- SVAMP: A benchmark consisting of simple arithmetic word problems created by perturbing examples from the well-known MAWPS dataset. It is designed to assess the robustness and true reasoning ability of language models by introducing variations that discourage rote memorization. Models must comprehend problem semantics and adapt to diverse formulations of similar mathematical tasks.

### HIGH SCHOOL LEVEL

- MATH: A dataset created by Dan Hendrycks et al., containing 12,500 problems from American high school math competitions (such as AMC 10, AMC 12, AIME). The problems cover multiple subjects including algebra, geometry, number theory, and counting & probability. Each problem includes a detailed solution written by a human expert in LaTeX format. Its difficulty is significantly higher than elementary school problems, making it a key benchmark for advanced mathematical reasoning.
- MMLU-STEM: MMLU includes 57 different subjects, and MMLU-STEM refers to the subset of subjects related to STEM (Science, Technology, Engineering, and Mathematics), such as college-level math, physics, chemistry, and computer science.
- Gaokao2023En: This dataset is derived from China's "Gaokao" (National College Entrance Examination) mathematics papers. It typically involves translating Chinese math problems into English to test a large model's ability to solve difficult math problems from different cultural and educational backgrounds.
- GaokaoMathQA: A high-school level benchmark constructed from authentic Chinese National College Entrance Examination (Gaokao) mathematics questions. The dataset includes both multiple-choice and open-ended problems covering a broad range of high-school topics, such as functions, geometry, probability, and calculus. It aims to evaluate models' abilities to perform symbolic reasoning and multi-step quantitative problem-solving under exam-style constraints.

### COLLEGE LEVEL & BEYOND

- CounterMath: A university-level mathematical benchmark designed to evaluate a model's conceptual reasoning by requiring it to prove or disprove statements by providing counterexamples. It focuses on advanced topics in Algebra, Topology, Real Analysis, and Functional Analysis.
- TheoremQA: A theorem-driven question answering dataset created to evaluate an AI model's ability to apply scientific theorems to solve challenging problems. It contains 800 questions covering over 350 theorems from Mathematics, Physics, EE&CS, and Finance.
- Minerva Math: A benchmark derived from the work on the Minerva model (Lewkowycz et al., 2022), which was designed to train large language models on scientific and mathematical content to enable step-by-step quantitative reasoning. The benchmark covers undergraduate-level math and science questions in natural language and LaTeX, requiring models to correctly parse, compute, and symbolically manipulate expressions without external tools.

### COMPETITION LEVEL

- Olympiad Bench: A benchmark of extremely challenging, Olympiad-level scientific problems in both mathematics and physics. It is designed to push the boundaries of AGI research and often includes multimodal elements, requiring models to interpret diagrams and perform complex, creative reasoning.

### B.3 MORE EVALUATION DETAILS

For evaluation, we adopt the SC@21 setting with a sampling temperature of 0.7. Specifically, 21 responses are generated, and after discarding cases where the answer cannot be extracted, the final

prediction is determined by majority voting. In the event of a tie, one of the tied candidates is selected uniformly at random.

## B.4 TRAINING PROMPT EXAMPLE

---

*Completion Prompt*

**System:** You are a helpful assistant that solves multiple-choice math questions with step-by-step reasoning.

**User:** Please solve the following question carefully. Explain your reasoning, and conclude with the final answer using the format: `\boxed{X}`, where X is A, B, C, or D.

Example:
Question: What is 2 + 3?
A. 4
B. 5
C. 6
D. 7

Answer: 2 + 3 = 5, which is option B.
The final answer is `\boxed{B}`.

—

Question: {specific math problem}
A. {option A}
B. {option B}
C. {option C}
D. {option D}

---

# C    ANALYSIS

## C.1    PPO-BASED VARIANT: OPTIMIZER-AGNOSTIC INSTANTIATION OF CORE

The CORE framework is designed as an architectural module that operates independently of the underlying reinforcement learning optimizer. While the primary experiments utilize GRPO for efficiency, CORE can be instantiated with other policy gradient methods, such as PPO, provided the structural requirement of multi-sampling is met. This section provides the technical details of the PPO-based implementation and its corresponding performance.

### ALGORITHMIC FORMULATION

To implement CORE within a PPO backbone, a batch configuration with $N = 4$ trajectories per prompt is utilized. To isolate the effects of the conceptual intervention and reduce reliance on additional neural components, the standard critic network is omitted. Instead, a group-wise Monte-Carlo baseline is employed for variance reduction. For each trajectory $i$, the discounted returns $G_i$ are first computed as follows:

$$G_i = \sum_{t'=t}^{T} \gamma^{t'-t} r_{t'} \tag{2}$$

The advantage $A_i$ is then estimated through group normalization of these returns within the sampled group:

$$A_i = \frac{G_i - \mathbb{E}_{\text{group}}[G]}{\text{std}_{\text{group}}(G)} \tag{3}$$

This formulation facilitates the integration of the CORE intervention logic, which encompasses both the Concept Bonus and Trajectory Replacement operations. Furthermore, such integration is achieved within the PPO clipped objective without necessitating any modifications to the core optimization step.

### EXPERIMENTAL RESULTS

As summarized in Table 6, the PPO-based instantiation of CORE achieves consistent performance gains across most mathematical benchmarks. These results highlight the modularity of the framework: since CORE reshapes the training distribution and reward signals at the data-processing level, it functions effectively as a higher-level supervision layer that is agnostic to the specific choice of the RL optimizer.

| Benchmark | Vanilla | CORE-CR (PPO) | Improvement |
|---|---|---|---|
| GK | 55.3 | **57.7** | +2.4% |
| MM | 29.8 | **30.1** | +0.3% |
| AM | 37.5 | **40.0** | +2.5% |
| CM | 37.3 | **43.2** | +5.9% |
| MS | 72.9 | **74.1** | +1.2% |
| MW | 96.8 | **97.5** | +0.7% |
| OL | 28.7 | **29.3** | +0.6% |
| AD | 95.1 | **95.2** | +0.1% |
| MH | 69.1 | **69.2** | +0.1% |

Table 6: Performance of the PPO-based variant under the SC@21 setting. Abbreviations: **AD**=ASDiv, **GK**=Gaokao 2023 (EN), **GS**=GSM8K, **MH**=MATH, **MM**=Minerva_math, **MS**=MMLU-STEM, **CM**=College_math, **AM**=Amc23, **MW**=MAWPS, **OL**=OlympiadBench.

## C.2    DISENTANGLING THE SOURCE OF CORE'S PERFORMANCE GAINS

To further disentangle the source of the performance gains achieved by CORE, we conduct a series of controlled ablation studies aimed at ruling out alternative explanations beyond our concept-guided training mechanism. In particular, we test two competing hypotheses:

Table 7: Ablation on **Qwen2-Math-7B** under SC@21 ($T$=0.7). We compare: *Vanilla*; *CORE-Base*; *Random-Reward GRPO*; *Top 4 of 6 GRPO*; and *CORE-CR*. Metrics are accuracy (%). Abbreviations: **AD**=ASDIV, **GK**=GAOKAO 2023 (EN), **GS**=GSM8K, **MW**=MAWPS, **OL**=OLYMPIADBENCH, **SV**=SVAMP, **TM**=TABMWP.

| Model | Method | AD | GK | GS | MW | OL | SV | TM |
|---|---|---|---|---|---|---|---|---|
| Qwen2-Math-7B | Vanilla | 95.1 | 55.3 | 89.8 | 96.8 | 28.7 | 92.5 | 90.2 |
| | CORE-Base | 95.4 | **59.5** | 90.8 | 97.2 | 33.9 | 92.1 | 92.6 |
| | Random-Reward GRPO | 95.2 | 55.8 | 89.9 | 97.2 | 29.9 | 91.2 | 89.6 |
| | Top 4 of 6 GRPO | 95.0 | 54.5 | 89.3 | **97.3** | 32.1 | 91.5 | 88.3 |
| | CORE-CR | **95.7** | 58.4 | **91.1** | 97.3 | **34.5** | **92.7** | **93.6** |

- **H1:** the gains are artifacts of the underlying GRPO optimization procedure.
- **H2:** the gains primarily arise from verifier-based process supervision.

Below, we evaluate each hypothesis through targeted control experiments.

**Testing H1: Do the Gains Come from CORE Rather than GRPO?** Next, we investigate whether the observed gains are simply artifacts of the GRPO optimization procedure, rather than a consequence of the concept-guided curriculum introduced by CORE. To this end, we perform controlled ablations on **Qwen2-Math-7B**, holding compute, data, and hyperparameters fixed, and evaluate all variants under the same SC@21 ($T$=0.7) protocol.

**Random-Reward GRPO.** For each generated response, we assign a binary reward at random (correct/incorrect) and train with GRPO on these randomized labels. This tests whether stochastic reward noise alone can yield apparent performance improvements.

**More-Candidates Top-$k$ Control.** Standard GRPO samples four responses per input. Here we additionally generate two extra candidates (six in total) and then select the four highest-reward responses for training. This controls for the effect of a larger rollout budget and stronger within-GRPO selection pressure, without introducing any concept-guided intervention.

**Results.** Table 7 shows that *Random-Reward GRPO* does not yield meaningful gains under the same setup. The *Top 4 of 6* variant produces only small and unstable changes, often mild regressions. This indicates that larger rollouts or stricter in-GRPO selection alone do not account for the improvements. By contrast, both *CORE-Base* and *CORE-CR* consistently outperform the vanilla baseline across benchmarks, with *CORE-CR* achieving the best overall performance.

**Testing H2: Process Supervision and Verifier-Guided RL.** As a first control, we compare CORE against a verifier-guided reinforcement learning variant, termed **Process Supervision and Verifier-Guided RL** and denoted as **CORE-Base + Verifier**. This baseline is designed to contrast explicit concept-guided intervention with passive process-level supervision, where learning signals are conditioned on intermediate reasoning trajectories rather than solely on final outcomes.

Concretely, a **Concept-Verifier** is integrated into the CORE-Base training loop, using the same **Qwen2-Math-7B** model as the verifier. For each generated reasoning trajectory, the verifier checks whether the target concept is explicitly and correctly invoked in the reasoning process. If the concept is correctly applied, an intrinsic process reward of $+0.4$ is assigned, independent of the final answer correctness. This verifier therefore serves as a sparse process-level reward model targeting conceptual alignment. All other hyperparameters are kept identical to those used in CORE.

Table 8 summarizes the performance of this verifier-guided RL baseline. While CORE-Base + Verifier yields modest gains over the vanilla model on several benchmarks, it consistently underperforms **CORE-CR**. This suggests that explicit concept-guided intervention during training is more effective than verifier-based process supervision alone, and that simply rewarding the presence of correct concepts in intermediate trajectories is insufficient to induce robust conceptual reasoning.

Overall, both control experiments consistently reject H1 and H2. Neither verifier-based process supervision nor GRPO artifacts alone can account for the observed improvements. The gains there-

Table 8: Comparison with a verifier-guided RL baseline under SC@21 ($T$=0.7). Columns use *two-letter* abbreviations: **GS**=GSM8K, **AD**=ASDiv, **MW**=MAWPS, **TM**=TabMWP, **MH**=MATH, **MS**=MMLU-STEM, **OL**=OlympiadBench.

| Model | Method | GS | AD | MW | TM | MH | MS | OL |
|---|---|---|---|---|---|---|---|---|
| | Vanilla | 89.8 | 95.1 | 96.8 | 90.2 | 69.1 | **72.9** | 28.7 |
| Qwen2-Math-7B | CORE-Base + Verifier | 90.8 | 94.5 | 96.0 | 92.5 | 69.7 | **72.9** | 34.2 |
| | CORE-CR | **91.1** | **95.7** | **97.3** | **93.6** | **71.4** | 72.6 | **34.5** |

fore primarily stem from the concept-guided curriculum and intervention mechanisms introduced by CORE.

## C.3 EXAMPLE FOR ANALYSIS

Below is an exmaple for analysis.

| CORE-Base | CORE-Base | CORE-KL |
|---|---|---|
| **Input:** Let $A$, $B$ be orthogonal matrices, $k$ be a nonzero real number, and $P$ an invertible matrix. Which of the following is true? A) $A + B$ is also an orthogonal matrix B) $kA$ is also an orthogonal matrix C) $AB$ is also an orthogonal matrix D) $P^{-1}AP$ is also an orthogonal matrix | **Input:** **Concepts:** Orthogonal and Unitary Transformations: Orthogonal and Unitary Matrices An $n$-dimensional real matrix $A$ is called an orthogonal matrix if $A'A = AA' = I_n$. An $n$-dimensional complex matrix $U$ is called a unitary matrix if $\overline{U}'U = U\overline{U}' = I_n$. ––––––––––– **Exercise:** Let $A$, $B$ be orthogonal matrices, $k$ be a nonzero real number, and $P$ an invertible matrix. Which of the following is true? A) $A + B$ is also an orthogonal matrix B) $kA$ is also an orthogonal matrix C) $AB$ is also an orthogonal matrix D) $P^{-1}AP$ is also an orthogonal matrix | **Input:** Let $A$, $B$ be orthogonal matrices, $k$ be a nonzero real number, and $P$ an invertible matrix. Which of the following is true? A) $A + B$ is also an orthogonal matrix B) $kA$ is also an orthogonal matrix C) $AB$ is also an orthogonal matrix D) $P^{-1}AP$ is also an orthogonal matrix |
| **Output:** Answer: C \| Response: B … B) $kA$ is also an orthogonal matrix. This is true. **If $A$ is an orthogonal matrix and $k$ is a nonzero real number, then $kA$ is also an orthogonal matrix. This is because multiplying an orthogonal matrix by a scalar does not change its orthogonality.** C) $AB$ is also an orthogonal matrix.This is true. If $A$ and $B$ are orthogonal matrices, then their product $AB$ is also an orthogonal matrix. This is because the product of two orthogonal matrices is orthogonal.… The final answer is boxed{B}. | **Output:** Answer: C \| Response: B … B) $kA$ is also an orthogonal matrix This is true. **If $A$ is an orthogonal matrix, then $kA$ is also an orthogonal matrix for any nonzero real number $k$.** C) $AB$ is also an orthogonal matrix This is true. If $A$ and $B$ are orthogonal matrices, then $AB$ is also an orthogonal matrix. Therefore, the correct answer is B) $kA$ is also an orthogonal matrix. | **Output:** Answer: C \| Response: C … B) $kA$ is not necessarily an orthogonal matrix. **If $k$ is a nonzero real number, then $kA$ is not orthogonal unless $k = pm1$. This is because the columns of $kA$ are not necessarily orthogonal unit vectors.** C) $AB$ is an orthogonal matrix. The product of two orthogonal matrices is also orthogonal. This is because the columns of $AB$ are linear combinations of the columns of $A$, which are orthogonal unit vectors. Therefore, the columns of $AB$ are also orthogonal unit vectors. … The correct answer is C) $AB$ is also an orthogonal matrix. The final answer is boxed{C}. |

Table 9: This is a case study comparing the original GRPO model, a variant augmented with concept prompts, and the model after **CORE-KL** training. The example illustrates that the original model tends to fall into a fixed reasoning pattern when dealing with orthogonal matrices. Even when provided with detailed conceptual derivations, the model still prefers to rely on patterns learned during pretraining. In contrast, the model trained with **CORE-KL** is able to break out of this fixed paradigm and effectively apply the relevant theorems of orthogonal matrices.

# D USE OF LLM

We have only used LLM for language polishing purposes in the paper writing. We do not use LLM for idea formalization, or to an extent that it could be regarded as a contributor.

