# OpenReview forum: "CORE: Concept-Oriented Reinforcement for Bridging the Definition–Application Gap in Mathematical Reasoning"
_ICLR.cc/2026/Conference — ICLR 2026 Poster_

### Official Review · Reviewer_b4Hh · 2025-10-21

**Soundness:** 2
**Presentation:** 3
**Contribution:** 2
**Rating:** 2
**Confidence:** 4

**Summary:**

This paper proposed the make a new math dataset that, in addition to question and answer pairs, augments related concepts by extracting and translating content from a Chinese math book. The authors then compared 3 GRPO based methods to show the effectiveness of their proposal.

Although the paper claims in the abstract that CORE is "an algorithm-agnostic training framework that turns explicit concepts ...", the fact that its algorithm relies on the $N$ generation failures suggests it is a derivation of GRPO, which invalidates the claim. In addition, I have 2 major concerns.

**Strengths:**

* The paper is tackling an important problem:
* While some details are hard to get (e.g., Sec 3.4), the paper is generally well-written.

**Weaknesses:**

* Unclear technical novelty of the RL method vs. the training data. While the paper emphasizes the goal of bridging the 'definition-application gap', its primary technical contribution (core-cr and core-kl) appears to be an application of existing RL strategies rather than a novel method. The core mechanism involves dynamically intervening with expert guidance when the policy fails. This is methodologically similar to prior work on "teacher" or "expert-anchored" RL. The paper's novelty therefore seems to be rest on the content of the intervention rather than the mechanism. But are the performance gains attributable to the unique properties of using a declarative concept, or would any high-quality reasoning trace produce the same benefit when used within this dynamic-intervention framework? The paper lacks an ablation study to disentangle the value of the content from the value of the method.
* Generalizability and conflation of methodological contribution with data curation. The method section describes a non-trivial, multi-stage curation pipeline to create the data, involving generations by Qwen2.5-72B-Instruct, vadilation and filtering by GPT4o, cross-model validation strategy, etc. This rigorous process suggests that the effectiveness of the proposed framework is linked to the quality fo the dataset. Actually, the core-base model, trained on this data, already shows significant improvement over the baselines. When comparing the performance boost after applying core-cr and core-kl, only TB and CM showed meaningful improvements. This thus questions if a large portion of the gains come from the data itself rather than the specific CORE-CR or CORE-KL intervention recipes.

**Questions:**

See details in the weaknesses section.

---

> ### Author Response · Authors · 2025-11-21
> **Rebuttal [1/3]**
>
> Thanks for your detailed review and constructive feedback. We appreciate you taking the time to provide such a thorough critique of our methodology and experimental setup, and we address your concerns point by point below.
>
>
> > Q1: It is a derivation of GRPO, which invalidates the claim
>
>
> ### A1
>
>
> We wish to emphasize that CORE is not proposed as a new RL optimization algorithm, but as a holistic framework for conceptual learning.
> CORE functions as a higher-level supervision module that systematically operationalizes textbook knowledge into training signals. Specifically, the CORE architecture comprises three integral stages, with GRPO optimization serving merely as the execution engine for the final stage (CORE-CR / CORE-KL) or for the last two stages in the case of CORE-Base:
>
> * **Curriculum Construction (Data-Level):** This foundational stage involves the rigorous selection of high-quality textbooks and the generation of quiz probes. This transforms static declarative knowledge into active evaluation protocols, establishing the necessary pedagogical scope.
> * **Concept-Oriented Supervision (Process-Level):** This stage generates the supervision signal upstream of the gradient update. During training, the model generates $N$ trajectories for a given prompt. In the CORE-Base setting, this multi-sampling serves as implicit consolidation, allowing the model to rehearse concept application through standard exploration. When CORE-CR or CORE-KL is employed, the framework further utilizes these groups to diagnose conceptual failures (all-wrong groups). Upon detection, it actively intervenes by injecting concept definitions and replacing failed trajectories, thereby reshaping the training distribution to correct specific deficits before the data reaches the optimizer.
> * **Policy Optimization (Optimization-Level):** Finally, the resulting trajectory data is fed into the RL optimizer to update the policy. This stage unifies inputs from both standard quiz exploration in CORE-Base and explicit concept injection in CORE-CR/KL. It functions as an execution backend. While our main results prioritize GRPO for efficiency, this design is inherently modular. As shown in our PPO experiments next, the framework is compatible with alternative optimization backbones.
>
>
> The only structural requirement for CORE is Multi-Sampling. This "group" does not need to be tied to a specific RL algorithm.
>
> * It can be achieved via parallel sampling in a batch (as in GRPO or standard PPO with multiple samples per question)
> * It could even be aggregated across training epochs or accumulation steps. As long as we can observe multiple attempts to identify consistent conceptual failure, **CORE’s** intervention logic applies. The subsequent **Concept Bonus** and **Trajectory Replacement** are flexible operations that function independently of how the Advantage is mathematically formulated.
>
> To explicitly demonstrate this modularity, we implemented **CORE** on top of a **PPO** backbone configured with multiple samples per question ($N=4$).
>
> * **Advantage Estimation (PPO with Group-Based Baseline):** We deliberately omitted the Critic network to isolate the variance-reduction mechanism. Instead of a learned value function, we utilize a group-wise Monte-Carlo baseline. Specifically, we first compute the discounted returns $G_t = \sum_{t'=t}^T \gamma^{t'-t} r_{t'}$ for each trajectory. Subsequently, we estimate the advantage by normalizing these returns using group statistics: $\displaystyle A\_i = \frac{G\_i - \mathbb{E}\_{\text{group}}[G]}{\text{std}\_{\text{group}}(G)}$
>
> * **Addressing the Similarity:** We acknowledge that this formulation shares mathematical similarities with GRPO’s outcome-based normalization. This is precisely the point: it demonstrates that "Group Normalization" is a generic variance-reduction technique compatible with PPO's clipped objective, independent of whether one uses scalar rewards or discounted returns.
>
> * **CORE's Independence:** Crucially, while we switched the optimizer to this **PPO-REINFORCE** formulation, the CORE intervention logic (Step 1 & 2 above) remained identical. We successfully applied the same concept-augmented reward shaping and trajectory replacement.
>
>
> This PPO-based instantiation yielded significant improvements as well:
>
> | Benchmark | Vanilla | CORE-CR (PPO) | Improvement |
> | :--- | :---: | :---: | :---: |
> | Gaokao2023EN | $55.3$ | $\mathbf{57.7}$ | $+2.4\\%$ |
> | Minerva_math | $29.8$ | $\mathbf{30.1}$ | $+0.3\\%$ |
> | Amc23 | $37.5$ | $\mathbf{40}$ | $+2.5\\%$ |
> | College_math | $37.3$ | $\mathbf{43.2}$ | $+5.9\\%$ |
> | MMLU-STEM | $72.9$ | $\mathbf{74.1}$ | $+1.2\\%$ |
> | MAWPS | $96.8$ | $\mathbf{97.5}$ | $+0.7\\%$ |
> | OlympiadBench | $28.7$ | $\mathbf{29.3}$ | $+0.6\\%$ |
> | ASDiv | $95.1$ | $\mathbf{95.2}$ | $+0.1\\%$ |
> | MATH | $69.1$ | $\mathbf{69.2}$ | $+0.1\\%$ |
> | GSM8K | $\mathbf{89.8}$ | $89.5$ | $-0.3\\%$ |
> > *Note: Results are reported under the same SC@21 setting.*

---

> ### Author Response · Authors · 2025-11-21
> **Rebuttal [2/3]**
>
> > Q2: Unclear technical novelty of the RL method vs. the training data. The approach is methodologically similar to prior work on “teacher” or “expert-anchored” RL. This methodological similarity, together with the rigorous data construction process, suggests that the effectiveness of the proposed framework is closely linked to the quality of the dataset.
>
>
> ### A2
>
>
> We suggest that CORE should be viewed as a joint contribution of both data and method. In other words, while many works can be viewed as "teacher or expert-anchored RL" from a high level, this alone does not diminish their contribution and novelty in how they achieve it, since it is far from trivial and solved. The quiz-probe generation stage and the subsequent training procedure are designed as one coherent pipeline, and they reinforce each other. This is precisely our contribution in this paper, which is an effective and novel framework, instead of an RL algorithm. Our goal is not only to introduce an optimization routine, but also to transform static textbook content into dynamic, tiered supervision signals that drive concept-level learning, in a manner inspired by how humans learn mathematics.
>
>
> Specifically, the process begins by transforming raw, passive textbook material into active evaluation protocols. By automating the extraction of declarative concepts and synthesizing aligned quiz probes, we create the prerequisite environment for concept-oriented learning. This leads to the first critical stage of our framework, which is CORE-Base. We emphasize that CORE-Base functions as far more than a mere comparative baseline. Instead, it represents the essential stage of implicit consolidation. Just as students first learn by solving exercises, CORE-Base forces the model to consolidate and restructure its understanding of concepts it already partially possesses. This process transforms passive, latent knowledge into an active, retrieval-ready state. It establishes the necessary cognitive scaffolding before any explicit intervention occurs.
>
>
> Building upon this foundation, we introduce the stage of explicit correction via CORE-CR/KL. Human learning involves targeted tutoring when standard practice fails, and our framework mimics this by identifying concepts that the model consistently fails to master during the consolidation phase. For these specific hard cases, CORE-CR/KL actively inject definitions and align the reasoning distribution. RQ1 and RQ2 (lines 370–446) provide a careful human analysis that reveals the deeper role of CORE-CR/KL: they encourage the model to explicitly invoke the appropriate concepts and enhance its robustness to spurious or irrelevant concepts.
>
>
> Regarding the source of our performance gains, we acknowledge that using Qwen2.5-72B-Instruct for quiz generation in our initial setup may give the impression that the improvements mainly come from a strong teacher model. To separate our contribution from the strength of the generator and to show that CORE does not rely on perfectly clean quiz data, we conduct the following “Self-Supervised” experiment:
>
> * **Generator & Learner:** We utilized Qwen-2-Math-7B-Instruct to generate 885 quizzes based on the 236 textbook concepts. The reason we don’t use Qwen2-Math-7B itself is that we found it is good at solving problems but cannot generate a large number of questions.
> * **Self-Verification:** To ensure quality without external supervision, we use Qwen-2-Math-7B to solve these quizzes using Self-Consistency (SC@21). We retained only those quizzes where the model's SC results matched its own generated ground truth, resulting in a subset of $670$ quizzes.
> * **Training:** We then applied **CORE-CR** to the Qwen-2-Math-7B model using this self-generated data. We trained for $3$ epochs while keeping other parameters consistent.
>
>
> As shown in the results below, CORE delivers consistent improvements across benchmarks even when the concept probes are self-generated.
>
>
> | Benchmark | Qwen2-Math-7B | **CORE-CR**(Self-Supervised)|
> | :--- | :---: | :---: |
> | GSM8K | $89.8\\%$ | $\mathbf{91.4\\%}$ |
> | ASDiv | $95.1\\%$ | $\mathbf{95.5\\%}$ |
> | MAWPS | $96.8\\%$ | $\mathbf{97.6\\%}$ |
> | TabMWP | $90.2\\%$ | $\mathbf{92.5\\%}$ |
> | MATH | $69.1\\%$ | $\mathbf{70.4\\%}$ |
> | MMLU-STEM | $72.9\\%$ | $\mathbf{73.1\\%}$ |
> | Gaokao2023EN | $55.3\\%$ | $\mathbf{57.7\\%}$ |
> | OlympiadBench | $28.7\\%$ | $\mathbf{32.9\\%}$ |
>
>
> It demonstrates that our improvements stem not from distilling the intelligence of a 72B model. At its core, CORE is driven by the quality of the textbook data itself. We primarily rely on the rigorous pedagogical structure and the step-by-step progression of concepts provided by these texts. With this work, we encourage the community to reconsider the importance of textbooks and to adopt training paradigms that more closely mirror how humans learn concepts, which could help us figure out the differences between human learning and LLM learning.

---

> ### Author Response · Authors · 2025-11-21
> **Rebuttal [3/3]**
>
> > Q3: Are the performance gains attributable to the unique properties of using a declarative concept, or would any high-quality reasoning trace produce the same benefit when used within this dynamic-intervention framework?
>
>
> ### A3
>
>
> The core idea of CORE is to mimic how humans rigorously learn mathematics from textbooks, which is fundamentally different from standard knowledge distillation. For this reason, we believe that comparing CORE-CR only against a baseline augmented with teacher traces is methodologically misaligned.
>
>
> Importantly, as shown by the “Self-Supervised” experiment in our response to Q2, CORE’s effectiveness is not strongly dependent on any external large model for data generation.
>
>
> By contrast, teacher-driven distillation naturally risks propagating imitation biases and inheriting the teacher model’s stochastic hallucinations [1]. By grounding our framework in axiomatically correct, human-verified textbooks, we explicitly aim to avoid these systematic errors. Our goal is to model the structured acquisition of knowledge, rather than simply imitate the statistical outputs of another LLM.
>
>
> [1] Gudibande, Arnav, et al. “The False Promise of Imitating Proprietary LLMs.” arXiv preprint arXiv: 2305.15717 (2023)

---

> > ### Comment · Reviewer_b4Hh · 2025-11-28
> >
> > Thank you for the responses. While A2 addresses my concern, A1 and A3 do not.
> >
> > For A1, the authors are technically correct in the sense that they demonstrated modularity by getting PPO to work. However, their PPO implementation uses "group-based variance reduction," which is mathematically very similar to the mechanism that makes GRPO work. While they proved it works with PPO, the "philosophy" of the method (generate $N$ samples, reinforce the best ones or replace bad ones) is still fundamentally very close to existing methods. This makes the paper over-claiming.
> >
> > For A3, what I'm asking is an ablation study: CORE with Concepts vs CORE with Generic CoT Traces, and the authors' argument from a philosophical standpoint ("we want to mimic humans") is an evasion.
> >
> > For these reasons, I'll keep my current rating.

---

> ### Author Response · Authors · 2025-11-30
> **Response to Reviewer b4Hh**
>
> Thanks for the reviewer's note on their remaining concerns: "similar to GRPO" and "no comparison with other distillation methods." We can fully address them, as we believe these concerns are from fundamental misconceptions of our contribution. We elaborate below.
>
>
> As we explained in A1 and A2, we never claim to propose a new RL algorithm, nor do we claim that CORE is a distillation method relying on a larger model’s reasoning. These are not presented as contributions of our work. Therefore it is entirely reasonable in our setup that we use GRPO as part of the overall algorithm.
>
>
> We also implement a PPO-based variant, which shows that CORE performs well with a different RL algorithm. The reviewer mentioned that the “philosophy” of our method (generating samples and reinforcing the best ones or replacing the bad ones) is fundamentally similar to existing approaches, and thus the paper is over-claiming. However, our claim is simply that CORE is agnostic to the underlying RL algorithm [line 95-96], and we provide empirical evidence that it works with both GRPO and PPO. The specific form of advantage estimation does not affect the implementation of CORE. We do not propose a new RL method in this paper, so we believe this concern about over-claiming may stem from a misunderstanding of our contributions.
>
>
> Additionally, we believe that our CORE-CR (Self-Supervised) experiment provides strong evidence that CORE does not rely on distilling reasoning signals from a larger external model (and the reviewer acknowledged that A2 addressed this concern). This already distinguishes CORE from mainstream “teacher-based” or “expert-anchored” RL approaches, which is why we do not include direct comparisons with distillation-based methods, because such comparisons would be meaningless regardless of whether those methods outperform or underperform. Also, our motivation is from the “split-brain syndrome” phenomenon in LLMs [1], i.e., the definition–application gap we study here. Our methodology directly tackles this problem, while CoT-based distillation has nothing to do with mitigating this issue. We aim to probe the true concept reasoning, instead of playing hypes with LLM-generated, not fully human-verified garbage.
>
>
> [1] Zhang, Zheng. “Comprehension Without Competence: Architectural Limits of LLMs in Symbolic Computation and Reasoning.” arXiv preprint arXiv: 2507.10624 (2025)

---

### Official Review · Reviewer_6tm1 · 2025-10-27

**Soundness:** 2
**Presentation:** 3
**Contribution:** 2
**Rating:** 6
**Confidence:** 4

**Summary:**

This paper introduces CORE, an RL training framework that turns explicit math concepts into a controllable supervision signal. The proposed approach comprises three key components: dataset curation, gap diagnostics, and a concept reinforcement recipe. The framework is evaluated on two 7B models across multiple mathematical reasoning benchmarks, demonstrating clear effectiveness under the given experimental setup.

**Strengths:**

1. The paper targets an important research problem: LLMs often solve math problems by pattern matching rather than genuine conceptual understanding

2. The proposed approach, including the curated dataset and the CORE-KL framework, demonstrates clear value and effectiveness.

**Weaknesses:**

1. My main concern lies in the experimental setup. Although the proposed method shows effectiveness under the reported setting, the evaluations are primarily conducted on relatively simple or medium-difficulty math benchmarks. Moreover, the baseline results (Table 2) are inconsistent with those reported in the corresponding technical reports. For example, the performance of Qwen2-Math-7B on GSM8K and MATH does not match, and it is unclear why the model performs significantly worse on the simpler GSM8K compared to the more challenging MATH.

2. The RL training set is synthesized and evaluated by LLMs, raising concerns about data quality and reliability. It remains unclear how the authors ensure the correctness of the training data and mitigate potential issues such as reward hacking when using LLMs as judges.

**Questions:**

See the weaknesses section

---

> ### Author Response · Authors · 2025-11-21
> **Rebuttal [1/3]**
>
> We sincerely thank the reviewer for your careful reading and constructive feedback, and for taking the time to provide such a detailed assessment of our work.
>
>
> > Q1: The evaluations are primarily conducted on relatively simple or medium-difficulty math benchmarks
>
>
> ### A1
>
>
> We respectfully note that our original evaluation suite already included challenging datasets such as OlympiadBench, TheoremQA, and CounterMath, which require competition-level or conceptual mathematical reasoning.
>
>
> At the same time, our scope is to demonstrate better understanding and application of the math concepts from the source data; as a result, much more challenging datasets than those we used would require a much larger search space (e.g., IMO) or much complicated math concepts (e.g., frontier math proofs); both are out of the scope of this paper. Nonetheless, to conclusively address the concern and demonstrate CORE's efficacy on complex, reasoning-intensive tasks, we provide a granular breakdown of our existing results and conduct additional evaluations on competition-level benchmarks.
>
>
> **1. Performance Breakdown on MATH Level 5 (The Hardest Subset)**
>
>
> We performed a stratified analysis on our reported MATH evaluation results to isolate the "Level 5" subset (N=1324). This subset represents the most difficult problems requiring complex multi-step reasoning.
> | Model | Accuracy |
> | :--- | :---: |
> | Qwen2-Math-7B | $69.41\\%$ |
> | CORE-Base | $71.90\\%$ |
> | **CORE-CR** | $\mathbf{72.36\\%}$ |
> | CORE-KL | $72.28\\%$ |
>
>
> On this sample of 1324 hard problems, CORE maintains a decisive lead over the baseline. This granular analysis confirms that the aggregate gains in Table 2 are not artifacts of performance on simple tasks. Instead, CORE demonstrates robust capability specifically on the most challenging subset of the MATH dataset.
>
>
> **2. New Evaluations on Competition Math (AIME 2024 & AMC 2023)**
>
>
> Complementing the stratified analysis, we conducted new experiments on two additional hard math competitions to stress-test the models at the absolute limit of their reasoning capabilities. We specifically selected AIME 2024 and AMC 2023
> | Model | AIME 2024 ($N=30$) | AMC 2023 ($N=40$) |
> | :--- | :---: | :---: |
> | Qwen2-Math-7B | $3.3\%$ | $37.5\%$ |
> | CORE-Base | $\mathbf{6.7\%}$ | $\mathbf{50\%}$ |
> | CORE-CR | $\mathbf{6.7\%}$ | $42.5\%$ |
> | CORE-KL | $\mathbf{6.7\%}$ | $47.5\%$ |
>
>
> Above results demonstrate that the performance improvements of CORE extend to challenging, high-difficulty benchmarks.
>
>
>
> > Q2: The baseline results (Table 2) are inconsistent with those reported in the corresponding technical reports
>
>
> ### A2
>
>
> Thank you for raising this point. The discrepancy arises from different decoding strategies used in the evaluations.
>
>
> The results in the Qwen2.5-Math technical report [1] are obtained using greedy decoding. In contrast, all our reported results in the paper are evaluated under a Self-Consistency (SC@21) setting: for each problem, we sample 21 candidate solutions and take a majority vote over the final answers. Consequently, the overall accuracies we report are naturally higher than the greedy-decoding numbers in the technical report.
>
>
> It is worth emphasizing that our evaluation code is directly adapted from the official Qwen2.5-Math repository [2]. We strictly preserve all evaluation details from that codebase, including benchmark-specific prompt templates and answer-extraction logic. The primary modification we introduced is the addition of the self-consistency mechanism to ensure a more robust evaluation of reasoning capabilities.
>
>
> [1] Yang, An, et al. “Qwen2.5-Math Technical Report.” arXiv preprint arXiv: 2409.12122 (2024)
>
>
> [2] Qwen2.5-Math Official Repository. https://github.com/QwenLM/Qwen2.5-Math

---

> ### Author Response · Authors · 2025-11-21
> **Rebuttal [2/3]**
>
> > Q3: It is unclear why the model performs significantly worse on the simpler GSM8K compared to the more challenging MATH
>
>
> ### A3
>
>
> We appreciate this insightful observation. The differential in gains between GSM8K and MATH is not an anomaly. Rather, it is an expected outcome driven by two structural factors: performance saturation and curriculum-task alignment.
>
>
> First, on elementary benchmarks like GSM8K [1], the Vanilla Qwen2-Math-7B baseline has already reached a saturation point of 89.8%. We argue that this high baseline is partially inflated by template memorization or overfitting during pre-training. This phenomenon has been highlighted in recent studies like GSM-Symbolic [2]. Moreover, as shown in the ReasonAgain paper [3], large amounts of GSM-like synthetic data can be easily generated programmatically.  Improving upon this memorization ceiling is exponentially difficult because the residual errors are often trivial calculation slips rather than conceptual failures. Despite this saturation, CORE still squeezes out a +1.3% gain.
>
>
> Second, and critically, the source of our supervision signal dictates where the improvements manifest. As detailed in Section 3.2, our framework is grounded in university-level algebra materials, which are intrinsically dense with abstract definitions and theorems. Consequently, there is a natural complexity mismatch between this high-level conceptual training and the grade-school arithmetic tested in GSM8K. Injecting abstract algebra definitions offers limited leverage for solving simple procedural word problems.
>
>
> In contrast, the difficulty and rigorous style of these university textbooks align far more closely with  challenging benchmarks like MATH [4] and OlympiadBench [5]. These tasks explicitly demand the understanding and application of abstract definitions—precisely the capability our method cultivates. This structural alignment is empirically confirmed by the difficulty scaling trend in our results. As the task difficulty increases, the benefit of CORE amplifies: we observe a modest +1.3% improvement on GSM8K, a stronger +2.6% on MATH, and a significant +5.8% on OlympiadBench.
>
>
>
> [1] Cobbe, Karl, et al. “Training verifiers to solve math word problems.” arXiv preprint arXiv: 2110.14168 (2021)
>
>
> [2] Mirzadeh, Iman, et al. “GSM-Symbolic: Understanding the Limitations of Mathematical Reasoning in Large Language Models.” arXiv preprint arXiv: 2410.05229 (2024)
>
>
> [3] Yu, Xiaodong, et al. “ReasonAgain: Using Extractable Symbolic Programs to Evaluate Mathematical Reasoning.” arXiv preprint arXiv: 2410.19056 (2024)
>
>
> [4] Hendrycks, Dan, et al. “Measuring Mathematical Problem Solving with the MATH Dataset.” arXiv preprint arXiv: 2103.03874 (2021)
>
>
> [5] He, Chaoqun, et al. “OlympiadBench: A Challenging Benchmark for Promoting AGI with Olympiad-Level Bilingual Multimodal Scientific Problems.” arXiv preprint arXiv: 2402.14008 (2024)

---

> ### Author Response · Authors · 2025-11-21
> **Rebuttal [3/3]**
>
> > Q4: The RL training data is synthesized and evaluated by LLMs, raising concerns about data quality, correctness, and the risk of reward hacking when using LLMs as judges.
>
>
> ### A4
>
>
> We appreciate the reviewer's scrutiny regarding data integrity. We wish to clarify two fundamental aspects of our pipeline that mitigate these concerns: the deterministic nature of our reward signal and the inherent robustness of the CORE framework.
>
>
> First, to reiterate, we explicitly clarify that our RL training procedure does not utilize an LLM-as-judge reward model during the online policy optimization phase. This distinction eliminates the risk of "reward hacking" typically associated with learned or heuristic verifiers. During training, the reward signal is derived exclusively from deterministic ground-truth matching. We extract the final answer from the model’s response using standard regular expressions and compare it directly with the ground-truth label provided in the dataset. Consequently, the optimization signal is binary and objective. The use of LLM-based verification is strictly limited to the offline data curation phase, where we employ a rigorous pipeline to filter out low-quality quizzes before training ever begins.
>
>
> Second, regarding the concern that synthetic data quality might be a bottleneck, we conducted a 'Self-Supervised' experiment. This serves to demonstrate that data quality or zero-error accuracy is not a strict prerequisite for the efficacy of CORE. We acknowledge that using Qwen-2.5-72B-Instruct in our initial experiments might ostensibly resemble distillation. To rigorously prove that CORE's gains are intrinsic to our methodology and robust to data imperfections, we implemented the following self-supervised protocol:
>
> * **Generator & Learner:** We utilized Qwen-2-Math-7B-Instruct to generate $885$ quizzes based on the $236$ textbook concepts. The reason we generally do not use Qwen2-Math-7B itself is that we found it is capable of solving problems but struggles to generate a large volume of diverse questions.
>
> * **Self-Verification:** To ensure quality without external supervision, we used Qwen-2-Math-7B to solve these quizzes using Self-Consistency (SC@21). We retained only those quizzes where the model's SC results matched its own generated ground truth, resulting in a refined subset of $670$ quizzes.
>
> * **Training:** We then applied **CORE-CR** to the Qwen-2-Math-7B model using this self-generated data. We trained for $3$ epochs while keeping other parameters consistent.
>
>
> As shown in the results below, CORE delivers consistent improvements across benchmarks even when the concept probes are self-generated.
>
>
> | Benchmark | Qwen2-Math-7B | **CORE-CR**(Self-Supervised) |
> | :--- | :---: | :---: |
> | GSM8K | $89.8\\%$ | $\mathbf{91.4\\%}$ |
> | ASDiv | $95.1\\%$ | $\mathbf{95.5\\%}$ |
> | MAWPS | $96.8\\%$ | $\mathbf{97.6\\%}$ |
> | TabMWP | $90.2\\%$ | $\mathbf{92.5\\%}$ |
> | MATH | $69.1\\%$ | $\mathbf{70.4\\%}$ |
> | MMLU-STEM | $72.9\\%$ | $\mathbf{73.1\\%}$ |
> | Gaokao2023EN | $55.3\\%$ | $\mathbf{57.7\\%}$ |
> | OlympiadBench | $28.7\\%$ | $\mathbf{32.9\\%}$ |
>
>
> Our “Self-Supervised” experiment provides clear evidence: even when we restrict ourselves to a generator of the same family and scale (Qwen2-Math-7B-Instruct) and do not perform heavy-quality filtering, CORE-CR still achieves substantial gains. This shows that the effectiveness of our framework does not depend on perfectly clean, error-free quiz data. Instead, the method is naturally robust and can extract useful learning signals even from noisy, self-generated supervision.
>
>
> At its core, CORE is driven by the quality of the textbook data itself. We primarily rely on the rigorous pedagogical structure and the step-by-step progression of concepts provided by these texts. With this work, we encourage the community to reconsider the importance of textbooks and to adopt training paradigms that more closely mirror how humans learn concepts, which could help us figure out the differences between human learning and LLM learning.

---

### Official Review · Reviewer_vWn1 · 2025-11-01

**Soundness:** 3
**Presentation:** 3
**Contribution:** 3
**Rating:** 6
**Confidence:** 3

**Summary:**

This paper presents concept-oriented approach to create synthetic data based on math concepts. The initial diagnostic evaluation results show that base models, while perform well in standard evaluation, they are prone to error when tested with variation inputs (that permutes parameters). The paper then presents an RL based approach to train models based on guidance from concepts in events of reasoning divergence. The results show that Core RL approaches can generalize beyond the synthetic puzzles.

**Strengths:**

- puzzle dataset construction based on math text book
- diagnostic analysis about issues of why current models could fail in truly understand math concepts.
- RL frameworks to train models to be coherent with concepts with different reward models
- evaluation shows out of domain generalization

**Weaknesses:**

- difference in RL methods CR / KL have some improvement over the baseline RL-base, it is not clear how these methods truly improve on top of baseline RL approach.
- (neutral comment) another simple baseline of enhancing models with variation data (e.g., with permuted params) could also be used to understand the wether concept reasoning is important, or it is just a matter of data augmentation is needed.

**Questions:**

Please elaborate comparison of three RL approaches, and how they differs in contribution to performance gains. It's good to understand importance of augmented data vs new RL methods.

---

> ### Author Response · Authors · 2025-11-21
> **Rebuttal [1/2]**
>
> We thank the reviewer for your thoughtful review, for acknowledging the strengths of our work, and for the constructive suggestions regarding the comparison of our RL approaches.
>
> > Q1: Please elaborate comparison of three RL approaches, and how they differ in contribution to performance gains.
>
> ### A1
>
> CORE-Base functions as a consolidation mechanism, enabling the model to review and reinforce the application of concepts that were already encountered during pre-training. In contrast, CORE-CR and CORE-KL serve as complementary corrective strategies designed for specific unmastered concepts. And we want to highlight that CR and KL are parallel, as the former explicitly changes the rollouts to have the policy imitate it, while the latter focuses on implicitly pushing the policy distribution to generate concept-based reasoning. These are two straightforward ways to incorporate the concepts into training. All three variants are our contributions, and we are just probing how concepts could be augmented differently.
>
>
> Next, we provide a detailed comparison of CORE-Base, CORE-CR, and CORE-KL to clarify their respective roles and differences.
>
>
> At a high level, CORE-Base works by converting textbook concepts into quiz-style probes, so that the model can consolidate and rehearse its understanding and application of each concept through reinforcement learning on these quizzes. In other words, CORE-Base mainly helps the model review concepts it is already somewhat familiar with by repeatedly solving concept-aligned questions under a standard GRPO setup.
>
>
> However, we observe that for a subset of quizzes, the model consistently fails even after training with CORE-Base. This typically happens when (1) the model essentially lacks the relevant concept, or (2) pretraining has pushed it into a “fixed pattern” of reasoning that it repeatedly falls back to despite negative feedback. To address these persistent failures, CORE-CR and CORE-KL introduce conditional interventions that explicitly inject the target concept into training exactly when the model fails, rather than relying only on outcome-based quiz rewards. Intuitively, while CORE-Base helps the model strengthen concepts it already partially knows, CORE-CR and CORE-KL are designed to teach the concepts it does not yet master.
>
>
> One direct indicator of their effect is quiz accuracy after training. After 3 epochs, we obtain:
> | Model | Accuracy | Raw Count (Correct / Total) |
> | :--- | :---: | :---: |
> | CORE-Base | $91.98\%$ | $1021 / 1110$ |
> | **CORE-CR** | $\mathbf{94.50\%}$ | $1049 / 1110$ |
> | CORE-KL | $94.32\%$ | $1047 / 1110$ |
>
>
> Thus, both CORE-CR and CORE-KL resolve a substantial portion of the quizzes that CORE-Base still gets wrong, consistent with their design goal of explicitly correcting concept-level weaknesses. In addition, as shown in Table 2, on almost all external benchmarks at least one of CORE-CR or CORE-KL outperforms CORE-Base, indicating that these concept-injection mechanisms translate into end-task gains, not just better quiz performance.
>
>
> We further conducted human analyses in RQ1 and RQ2 (Section 5.1 and 5.2, line 370-446) to understand how CORE-CR/KL improve the model’s reasoning:
>
>
> Mechanism Shift (RQ1): In our audit of difficult cases where Vanilla/CORE-Base failed but CORE-CR&KL succeeded, we found zero instances of shortcut learning. In 100% of these cases, CORE-CR/KL succeeded by explicitly invoking the correct concept. This proves that while CORE-Base relies on the model's internal retrieval, CORE-CR/KL contributes by actively installing the correct reasoning path where the internal mechanism falters.
>
>
> Robustness (RQ2): As shown in Figure 2, CORE-CR/KL significantly outperform CORE-Base in resisting irrelevant distractor concepts (maintaining a >10% lead). This indicates that explicit intervention contributes not just to accuracy, but to the stability of the reasoning process.
>
>
> These clarify how CORE-CR/KL build on and surpass the baseline RL approach embodied by CORE-Base. CORE-Base provides the necessary breadth by consolidating known concepts, while CORE-CR/KL provide the critical depth by repairing specific conceptual deficits and enforcing robustness.

---

> ### Author Response · Authors · 2025-11-21
> **Rebuttal [2/2]**
>
> > Q2: (neutral comment) Another simple baseline of enhancing models with variation data (e.g., with permuted params) could also be used to understand whether concept reasoning is important, or it is just a matter of data augmentation is needed.
>
>
>
> ### A2
>
>
>
> We thank the reviewer for raising this interesting and novel suggestion. However, we think that this additional baseline cannot be carried out and is also unnecessary, given the nature of our concept quizzes. Our quizzes are designed primarily to assess the model’s understanding of concepts rather than its computational ability, so there is inherently limited room for meaningful variation-based augmentation. Even if one modifies the training data through such variations, the core requirement of correctly applying the underlying concepts remains unchanged. Moreover, evaluating the model solely by adding more variant data and checking whether benchmark scores increase would deviate from our original goal, which is to probe and improve conceptual understanding rather than to perform generic data augmentation.

---

### Official Review · Reviewer_kr6a · 2025-11-01

**Soundness:** 3
**Presentation:** 3
**Contribution:** 2
**Rating:** 6
**Confidence:** 3

**Summary:**

This paper focuses on the definition–application gap in large language models (LLMs) for mathematical reasoning, ie, models can accurately recite formal definitions but often fail to select and apply the correct concepts when solving problems.

To address this issue, the authors curate a structured concept–exercise corpus from a linear algebra textbook, containing 236 concept definitions, 703 examples, and 140 exercises. They further construct a Concept Probes dataset of 1,110 multiple-choice quizzes by generating items with Qwen2.5-72B and filtering them through GPT-4o for quality assurance. This dataset serves both diagnostic and training purposes.

Building on GRPO reinforcement learning, the authors propose CORE (Concept-Oriented REinforcement) through three variants:

1. CORE-Base: standard RL training directly on the concept-probe dataset.

2. CORE-CR: when all sampled trajectories fail, failed ones are partially replaced with concept-guided rollouts and assigned a reward bonus.

3. CORE-KL: upon failure, the model minimizes the forward KL divergence between the concept-guided and unguided policies, effectively distilling the reasoning distribution that emerges when a concept is provided.

Experiments on Qwen2-Math-7B and Llama-3-8B-Instruct show that all CORE variants consistently improve both in-domain textbook exercises and multiple out-of-domain mathematical reasoning benchmarks, demonstrating enhanced concept selection and application.

**Strengths:**

The paper introduces concept texts as a controllable and fine-grained supervision signal in reinforcement learning, offering richer and more directed feedback compared to traditional scalar rewards based solely on final answer correctness.


Experiments are conducted on both base and instruction-tuned models across multiple, stylistically diverse mathematical benchmarks, demonstrating the generality and robustness of the proposed approach.


Through small-scale manual case analysis and perturbation tests with irrelevant concepts, the authors provide preliminary evidence that the performance gains stem from improved concept selection and application rather than superficial heuristic matching.

**Weaknesses:**

The comparison with relevant baselines is insufficient: the paper does not include fair experimental comparisons with conceptually similar methods such as BREAD (which also employs failure-branch replacement), nor with other related process-supervision or verifier-guided RL approaches. As a result, the marginal contribution and uniqueness of CORE remain unclear.

Most reported performance improvements are relatively small and lack statistical significance.

**Questions:**

See Weakness.

---

> ### Author Response · Authors · 2025-11-21
> **Rebuttal [1/2]**
>
> We sincerely thank the reviewer for your careful reading and constructive feedback.
>
> > Q1: The comparison with relevant baselines is insufficient: the paper does not include fair experimental comparisons with conceptually similar methods such as BREAD (which also employs failure-branch replacement), nor with other related process-supervision or verifier-guided RL approaches. As a result, the marginal contribution and uniqueness of CORE remain unclear.
>
> ### A1
>
>
> **Comparison with Teacher-Distilled Trajectory Replacement (e.g., BREAD)**
>
>
> We thank the reviewer for highlighting BREAD and related trajectory replacement works. Regarding a direct empirical comparison, to the best of our knowledge, the official implementation for BREAD has not yet been open-sourced. We are committed to including a rigorous comparison in the final version of the paper as soon as their code becomes available or reproduction is feasible.
>
>
> In addition, while we acknowledge technical similarities in the mechanism of replacing trajectories, we emphasize that CORE represents a fundamentally different research paradigm in terms of motivation, data source, and learning objective.
> BREAD and similar methods primarily aim to distill high-quality training data from stronger "teacher models" to improve a target model. Their focus is on correctness and process mimicry. Effectively, they act as data augmentation techniques to bridge the gap between SFT and RL.
>
>
> In contrast, CORE is motivated by the human cognitive process of learning mathematics.  This point of departure is fundamentally different from that of LLM distillation-based approaches. Our goal is not just to provide a "correct trace" but to force the model to anchor its reasoning in specific, declarative definitions. This alignment between definition and application is a distinct objective that standard distillation does not explicitly target. In addition, our method only asks models to generate grounded quiz questions instead of high-quality reasoning traces, so we do not rely on distillation from strong models. As we will show in the final version, using a 7B model to generate the quiz, CORE can still achieve comparable performance gains.
>
>
> By grounding our framework in human-verified textbooks, CORE circumvents the “harvester bias” and the potential hallucinations inherent in teacher-distilled trajectories. We aim to model the cognitive process of human learning, in which concepts are systematically acquired and consolidated both prior to and during application.
>
>
> **Comparison with Process Supervision and Verifier-Guided RL**
>
>
> To benchmark against process supervision, we implemented a Verifier-Guided RL baseline, denoted as CORE-Base + Verifier. Serving as a baseline for process supervision [1, 2], this method conditions the reward signal on the content of the reasoning trajectory itself, rather than solely on the final outcome. Specifically, we integrated a Concept-Verifier into the CORE-Base training loop. We employed the Qwen2-Math-7B model itself as a verifier. For each generated trajectory, the verifier checks if the reasoning explicitly and correctly invokes the target concept. If the concept is correctly applied, we assign an intrinsic process reward of +0.4, independent of the final answer correctness. This acts as a sparse process reward model targeting conceptual alignment. All other hyperparameters remain identical to CORE. We have run this verifier-guided RL baseline and summarize its performance below, so that CORE can be directly compared against this method：
> | Model                 | GSM8K (\%) | ASDiv (\%) | MAWPS (\%) | TabMWP (\%) | MATH (\%) | MMLU-STEM (\%) | OlympiadBench (\%) |
> |-----------------------|----------:|----------:|-----------:|------------:|----------:|---------------:|-------------------:|
> | Qwen2-Math-7B         | 89.8      | 95.1      | 96.8       | 90.2        | 69.1      | **72.9**       | 28.7              |
> | CORE-Base + Verifier  | 90.8      | 94.5      | 96.0       | 92.5        | 69.7      | **72.9**       | 34.2              |
> | CORE-CR               | **91.1**  | **95.7**  | **97.3**   | **93.6**    | **71.4**  | 72.6           | **34.5**          |
>
> We observe that introducing the CORE-Base + Verifier baseline yields modest gains over the vanilla model on some benchmarks, but it consistently falls short of CORE-CR.
>
>
> We hope this response effectively delineates CORE from 'Teacher-Distilled Trajectory Replacement' and the broader class of process-supervision or verifier-guided RL methods. By prioritizing Concept Alignment over simple procedural mimicry, CORE addresses a fundamental reasoning gap that these paradigms do not explicitly target.
>
>
> [1] Uesato, Jonathan, et al. “Solving Math Word Problems with Process- and Outcome-Based Feedback.” arXiv preprint arXiv: 2211.14275 (2022)
>
>
> [2] Lightman, Hunter, et al. “Let's Verify Step by Step.” arXiv preprint arXiv: 2305.20050 (2023)

---

> ### Author Response · Authors · 2025-11-21
> **Rebuttal [2/2]**
>
> > Q2: Most reported performance improvements are relatively small and lack statistical significance.
>
> ### A2
>
> ### Analysis of Statistical Significance
>
> We appreciate the reviewer's rigorous focus on statistical validity. While individual benchmark gains might appear modest in isolation, a holistic analysis reveals that the improvements are statistically significant, systematic, and amplified by difficulty.
>
>
> To provide a comprehensive picture, we first aggregated all eight main math benchmarks into a single cumulative metric, AVG@ALL. On this massive test set comprising 11489 instances, CORE demonstrates a clear and robust lead over the baseline:
> | Model | Accuracy |
> | :--- | :---: |
> | Qwen2-Math-7B | $79.34\%$ |
> | **CORE-CR** | $\mathbf{81.16\%}$ |
>
> We emphasize that the statistical validity of these results is underpinned by the sheer scale of our evaluation. Furthermore, conducting all evaluations with SC@21 substantially mitigates the variance associated with decoding randomness compared to greedy sampling. When combined with the consistent trajectory of improvement across different methods, these results demonstrate that CORE’s performance advantages are statistically robust and systematic, rather than artifacts of stochastic chance.
>
> We next report statistical significance results. Below is a clear and statistically significant performance improvement of **CORE-CR** over the **Vanilla** model on the specified domains ($N = 11489$).
>
> * **Accuracy:** $81.16\%$ (CORE-CR) vs. $79.34\%$ (Vanilla) $\rightarrow \Delta = +1.82$ percentage points
> * **McNemar’s test (paired, exact):** $n_{10} = 466$, $n_{01} = 257$
> * **Paired 95% Confidence Interval for $\Delta$ (bootstrap):** $[+1.36, +2.28]$ pp
> * **Discordant rate:** $(466 + 257) / 11489 = 6.3\\%$ of items differ between the two systems
> * **Paired odds ratio:** $n_{10} / n_{01} = 1.81$ with ~95% CI $[1.56, 2.11]$$\rightarrow$ On items where the two models disagree, CORE-CR is about $1.8\times$ more likely to be correct than the Vanilla model ($466$ wins vs. $257$), indicating a clear and systematic advantage rather than random fluctuation.
>
> Although the absolute accuracy gain is modest, the effect is highly statistically reliable. The disagreement cases are strongly skewed in favor of CORE-CR, indicating a consistent advantage rather than random fluctuation. We assess significance using a paired design via McNemar’s exact test on per-item correctness over the same 11489 samples, and quantify uncertainty using a paired bootstrap 95% confidence interval. This confirms that the approximately 1–2 percentage-point improvements of CORE-Base and CORE-CR over the baseline are statistically reliable.
>
>
>
> ### Clarification on Performance Magnitude
>
> To specifically address the concern regarding "relatively small" gains, we further stratified 10 benchmarks into three difficulty levels: Level 1 (GSM8K, ASDiv, MAWPS, TabMWP), Level 2 (MATH, MMLU-STEM, Gaokao2023En), and Level 3 (OlympiadBench, Textbook, TheoremQA). This stratification reveals that the aggregate improvement is masked by performance saturation on easier tasks:
>
> | Level | Qwen2-Math-7B | CORE-Base | **CORE-CR** | CORE-KL |
> | :--- | :---: | :---: | :---: | :---: |
> | **Level 1** ($N=6599$) | $93.8\%$ | $94.6\%$ | $\mathbf{94.9\%}$ | $94.4\%$ |
> | **Level 2** ($N=8403$) | $69.8\%$ | $71.2\%$ | $\mathbf{71.2\%}$ | $70.9\%$ |
> | **Level 3** ($N=867$) | $31.8\%$ | $36.9\%$ | $\mathbf{37.7\%}$ | $37.3\%$ |
>
>
> The perception of modest gains is largely an artifact of performance saturation on Level 1 benchmarks, where baselines already approach 94%. In stark contrast, on Level 3 benchmarks where conceptual reasoning is crucial, CORE-CR delivers a remarkable relative improvement. This pattern suggests that CORE is particularly effective for complex, reasoning-heavy tasks that necessitate deep conceptual understanding. We also attribute this in part to the advanced level of our selected textbook, which naturally aligns with the demands of these challenging benchmarks.

---

### Author Response · Authors · 2025-11-25
**Summary of the Discussion and Appreciation for Reviewers and Area Chairs**

Dear Reviewers and Area Chair,


As the discussion phase has now passed its two-thirds point, we would like to sincerely thank you for the time and care you have dedicated to our submission. Your comments and suggestions have been very helpful in refining both the ideas and the presentation of this work. We look forward to hearing your thoughts on our rebuttals. We deeply value your guidance and stand ready to answer any remaining questions or concerns.

**Main Contributions**
* **Bridging the Gap:** Our paper investigates the definition–application gap in mathematical reasoning and introduces CORE, a reinforcement learning framework that transforms textbook concepts into controllable supervision signals.
* **Beyond Scaling & Drills:** Unlike traditional "drill-based" training or the prevailing trend of simply scaling up synthetic data, CORE establishes a novel "Textbook-Oriented" training paradigm. It achieves stable and significant performance gains with minimal data and limited training resources.
* **Cognitive Inspiration:** By training models to mimic the human process of learning and internalizing mathematical concepts, we believe this paradigm offers rich, pedagogical insights and opens new directions for the future training of mathematical LLMs.

**Addressed Concerns**


During the rebuttal phase, we have strived to address all raised concerns with detailed clarifications and extensive new experiments:
* **Decoupling Method from Data Quality:** We added experiments to decouple the effectiveness of CORE from the quality of the training data. This clearly distinguishes our method from mainstream distillation approaches, proving the gains stem from the intrinsic concept mechanism.
* **Algorithmic Generalizability:** We demonstrated that CORE is algorithm-agnostic and compatible with a wide range of RL algorithms, including both GRPO and PPO.
* **Clarifying Variants & Inspiration:** We provided a comprehensive comparison of the three CORE variants to clarify their distinct roles and elaborated on how our framework is grounded in the cognitive science of human textbook learning.
* **Rigorous Analysis:** We incorporated a new baseline (Verifier-Guided RL) and conducted a more granular analysis of CORE’s improvements across different benchmarks.


All these updates will be carefully integrated into the final version of the paper. We also confirm that, if accepted, we will release our code. We deeply appreciate your attention to our work and the pivotal role you play in maintaining the high standards of ICLR. Thank you once again for your detailed assessment and constructive guidance throughout this process.


Best regards,


Authors of Submission 10456

---

### Meta-Review · Area_Chair_KirT · 2026-01-18

**Summary:**

This paper proposes CORE, an RL framework that turns explicit concepts into a controllable supervision signal. Experiments on 7B models show consistent improvements on both in-domain textbook exercises and a wide range of out-of-domain math benchmarks. Reviewers generally appreciate the problem formulation, the use of concepts as fine-grained supervision, and the breadth of evaluation. I recommend acceptance.

**Reviewer Concerns:**

Reviewer kr6a raised a concern about insufficient comparison with relevant baselines. In the rebuttal, the authors clarified the conceptual distinction between CORE and BREAD, and added a direct comparison with a Verifier-Guided RL baseline.

Reviewer kr6a also questioned whether the reported gains were statistically significant, and the authors addressed this by providing a thorough statistical analysis with paired tests and aggregated evaluations.

Reviewer vWn1 asked for a clearer comparison among CORE-Base, CORE-CR, and CORE-KL. The authors clarified the distinct roles of the three variants and supported this with additional analyses on quiz accuracy.

Reviewer 6tm1 raised concerns that the evaluation focused mainly on simple benchmarks. The authors clarified that hard benchmarks were already included and added new competition-level evaluations.

Reviewer 6tm1 also noted that the RL training set is synthesized and evaluated by LLMs, raising concerns about data quality and reliability. The authors explained that the RL reward is deterministic, thus avoiding reward hacking.

Reviewer b4Hh argued that the proposed RL methods do not have technical novelty, and vanilla RL on their proposed dataset works almost as well. In the rebuttal, the authors clarify that CORE should be viewed as a joint contribution of both data and method, and the core idea of CORE, which is to inject explicit concept signals into rollouts, is largely independent of the specific RL algorithm used.

Reviewer b4Hh also argued that similar performance gains might be achievable by training on generic high-quality reasoning traces. While this concern may remain after the rebuttal, the authors provided additional analyses and experiments with self-generated data to disentangle data quality from the core concept-injection mechanism.

**Reviewer Scores:**

* Reviewer kr6a is likely to keep the positive score 6, as the rebuttal addresses the concerns on baseline comparisons and statistical significance, while some reservations remain regarding the marginal magnitude of the performance gains.
* Reviewer vWn1 is likely to increase the score to 8, since the authors clarified the differences among the proposed methods and no major concerns remain.
* Reviewer 6tm1 is likely to increase the score to 8, as the added competition-level evaluations and clarification on deterministic rewards should have addressed the concerns.
* Reviewer b4Hh is likely to keep the score 2.

---

### Decision · Program_Chairs · 2026-01-26

Accept (Poster)